# A H$_2$O$_2$-activatable nanoprobe for diagnosing interstitial cystitis and liver ischemia-reperfusion injury via multispectral optoacoustic tomography and NIR-II fluorescent imaging

Junjie Chen[1,3], Longqi Chen[1,3], Yinglong Wu[2], Yichang Fang[1], Fang Zeng[1✉], Shuizhu Wu [1✉] & Yanli Zhao [2✉]

Developing high-quality NIR-II fluorophores (emission in 1000–1700 nm) for in vivo imaging is of great significance. Benzothiadiazole-core fluorophores are an important class of NIR-II dyes, yet ongoing limitations such as aggregation-caused quenching in aqueous milieu and non-activatable response are still major obstacles for their biological applications. Here, we devise an activatable nanoprobe to address these limitations. A molecular probe named BTPE-NO$_2$ is synthesized by linking a benzothiadiazole core with two tetraphenylene groups serving as hydrophobic molecular rotors, followed by incorporating two nitrophenyloxoacetamide units at both ends of the core as recognition moieties and fluorescence quenchers. An FDA-approved amphiphilic polymer Pluronic F127 is then employed to encapsulate the molecular BTPE-NO$_2$ to render the nanoprobe BTPE-NO$_2$@F127. The pathological levels of H$_2$O$_2$ in the disease sites cleave the nitrophenyloxoacetamide groups and activate the probe, thereby generating strong fluorescent emission (950~1200 nm) and ultrasound signal for multi-mode imaging of inflammatory diseases. The nanoprobe can therefore function as a robust tool for detecting and imaging the disease sites with NIR-II fluorescent and multispectral optoacoustic tomography (MSOT) imaging. Moreover, the three-dimensional MSOT images can be obtained for visualizing and locating the disease foci.

---

[1] Biomedical Division, State Key Laboratory of Luminescent Materials and Devices, Guangdong Provincial Key Laboratory of Luminescence from Molecular Aggregates, College of Materials Science and Engineering, South China University of Technology, Wushan Road 381, Guangzhou 510640, China. [2] Division of Chemistry and Biological Chemistry, School of Physical and Mathematical Sciences, Nanyang Technological University, 21 Nanyang Link, Singapore 637371, Singapore. [3] These authors contributed equally: Junjie Chen, Longqi Chen. ✉email: mcfzeng@scut.edu.cn; shzhwu@scut.edu.cn; zhaoyanli@ntu.edu.sg

Fluorescence imaging technique holds a great promise as an intuitional and effectual way to track and understand various biological and physiological processes because of its advantages such as high sensitivity and real-time manner. Fluorescent imaging in the second biological window (emission in 1000–1700 nm, NIR-II) has attracted increasing interests due to its immanent advantages including low light scattering and weak autofluorescence in tissues, which enables real-time imaging at much deeper depth with higher resolution in organism and could be employed for more accurate diagnosis of disease, monitoring of therapy course and understanding disease development[1–4]. Therefore, developing high-quality NIR-II fluorophores for in vivo imaging is of great significance and has a major impact on the field of preclinical and clinical applications.

Among the NIR-II organic fluorescent dyes, those based on benzothiadiazole backbone have aroused extensive attention because of the ease of extending their fluorescent emission into the NIR-II range, excellent photostability and large stoke shifts[5–8]. For NIR-II fluorescent dyes based on benzothiadiazole, the most common molecular structure is the donor–acceptor–donor (D-A-D) structure with the electron-accepting benzothiadiazole as the core and hydrophobic aryls as the electron-donating groups at the both sides of the core[9–12]. The integration of the two electronic donors (D) and an electronic acceptor (A, benzothiadiazole) is conducive to realizing a large and planar conjugation architecture, and thus effectively extending the fluorophore's emission to the NIR-II range[13–16]. Prime limitations associated with such large and planar conjugation include poor water solubility and the accompanying fluorescence quenching because of the aggregation in aqueous medium, and these limitations have become the major obstacles for exploiting these types of fluorophores in biological applications.

To overcome the aggregation-caused quenching (ACQ) in aqueous milieu for benzothiadiazole-core NIR-II fluorophores, the aggregation-induced emission (AIE) feature has been employed in the design of the molecular structure[17–22]. Unlike common fluorophores for which aggregation quenches their fluorescent emission, AIE-active fluorophores exhibit even stronger fluorescent emission as the aggregation extent is enhanced. In the AIE-active fluorophores, the molecular rotors dissipate the excited energy via non-radiative pathway by means of intramolecular movements such as rotations when the fluorophores are in the molecularly-dissolved form, whereas if the fluorophores are in the aggregation form, the movements of the rotors are frozen and the excited energy dissipates via radiative pathway and subsequently produces strong fluorescent emission[23–27]. Some hydrophobic molecular rotors such as triphenylamine and tetraphenylethylene (TPE) and their derivatives have often been employed as the donor groups at the both sides of benzothiadiazole core to make NIR-II fluorophores for bioimaging[28–30]. By this way, such disadvantage as poor water-solubility could be turned into an advantageous asset for devising NIR-II fluorophores on the strength of AIE. However, for benzothiadiazole-core NIR-II fluorophores, another major limitation still needs to be circumvented, namely, these fluorophores are generally inert probes which cannot respond to specific stimuli to achieve biomarker-activated detection or imaging. Inert probes give out unvaried or "always-on" signals that constitute the added background noise, while activatable probes produce signals only if they encounter or react with the target biomarker, hence the detection or imaging using activatable probes would have much higher sensitivity with negligible background noise and could effectively avoid giving false positive signals[31–36]. If only AIE-active NIR-II fluorophores based on benzothiadiazole core can be converted into biomarker-activatable probes, the probes would perceptibly hold a great potential for more extensive applications including clinical translation.

Optoacoustic (photoacoustic) imaging is an emerging modality that collects ultrasound waves generated by photoexciting contrast agents in tissues and produces images with high resolution and penetration depth[37–44]. A dual-mode probe applicable in both NIR-II fluorescence and optoacoustic imaging is highly advantageous because it provides mutually corroborated information. Reactive oxygen species (ROS) play essential roles in the progression of various inflammatory diseases such as drug-induced liver injury, liver ischemia/reperfusion injury and interstitial cystitis. At the foci of these diseases, the elevated level of $H_2O_2$ and other ROS lead to the intensified oxidative stress and subsequently cause tissue injuries and exacerbate inflammation. Hence endogenous hydrogen peroxide can serve as a promising in situ biomarker for these diseases. As ROS has a very short half-life (e.g., a few seconds and less), the in situ and non-invasive detection of the disease marker would be ideal for accurately locating the disease sites with the aid of robust probes via NIR-II fluorescent imaging and optoacoustic imaging.

In light of the above and endeavoring to devise an activatable two-mode NIR-II probe for disease diagnosis in a sensitive and non-invasive way, we developed a de novo nanoprobe namely BTPE-NO$_2$@F127 (Fig. 1). To obtain the nanoprobe, a benzothiadiazole-based core (compound 6, as shown in Supplementary Fig. 1) is first synthesized by connecting benzothiadiazole with two TPE, which is then linked with two nitrophenyloxoacetamide groups at its both ends to afford the molecular probe BTPE-NO$_2$. The molecular probe (BTPE-NO$_2$) is then encapsulated with the FDA-approved amphiphilic and biocompatible polymer Pluronic F127 (poly(ethylene oxide)-block-poly(propylene oxide)-block-poly(ethylene oxide)), affording the nanoprobe BTPE-NO$_2$@F127.

In this work, the design strategy for the probe BTPE-NO$_2$ lies in the following three aspects: (1) The two TPE groups serve as the hydrophobic molecular rotors, so as to make the chromophore BTPE-NH$_2$ AIE-active and simultaneously enhance the aggregation extent due to the increased hydrophobicity and conjugation; (2) The two nitrophenyloxoacetamide groups on both ends of the benzothiadiazole core serve not only as the recognition moiety for the biomarker $H_2O_2$, but also as an emission quencher owing to their electron-withdrawing ability; (3) Encapsulation of molecular probe BTPE-NO$_2$ by Pluronic F127 ensures necessary biocompatibility and water-dispersibility for biological applications. Importantly, owing to existence of the strong electron-withdrawing nitro group (NO$_2$) on the recognition moiety, the electronic state of the molecular probe can be significantly altered after the recognition moiety being cleaved. Without the presence of $H_2O_2$, this nanoprobe is almost non-fluorescent due to the existence of the two fluorescence quenchers and the absorption is centering around 615 nm, whereas the $H_2O_2$ at pathological level in the disease sites (e.g., in liver or bladder) cleaves the nitrophenyloxoacetamide and produces the activated chromophore (BTPE-NH$_2$), thereby red-shifting the absorption band to 680–850 nm and generating strong NIR-II fluorescent emission in the range of 950~1200 nm.

## Results

**Preparation and characterization of the nanoprobe BTPE-NO$_2$@F127.** The nanoprobe BTPE-NO$_2$@F127 was employed in the diagnosis of three inflammatory diseases in mouse models, including the trazodone-induced liver injury, the liver ischemia/reperfusion injury, and the interstitial cystitis, so as to evaluate its capability for detecting and imaging inflammatory disease sites via responding to the in situ endogenous biomarker. Our results perceptibly indicate that the nanoprobe BTPE-NO$_2$@F127 can function as a tool for biomarker-activated detection and imaging of

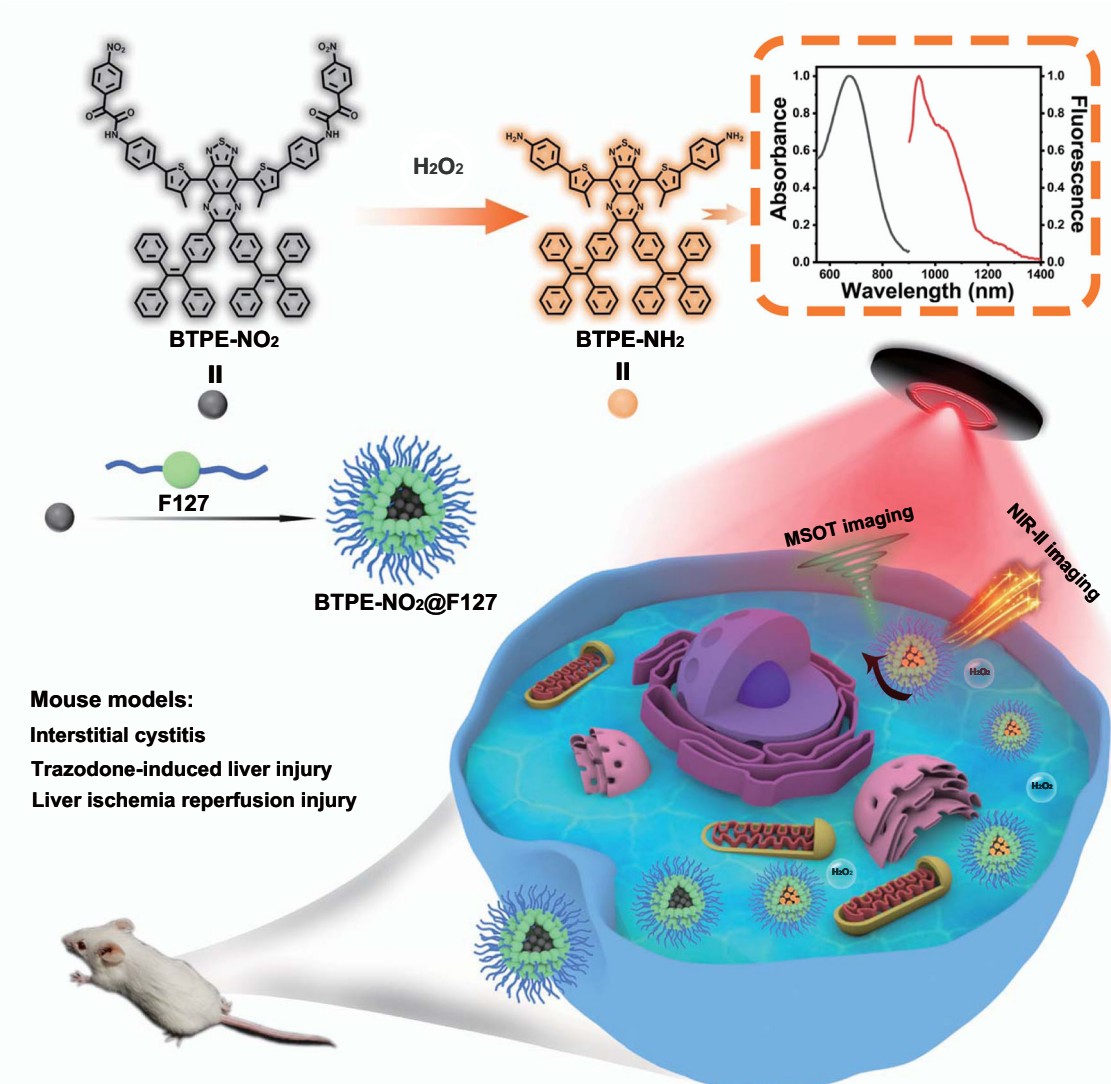

**Fig. 1 Schematic representation for the fabrication of nanoprobe BTPE-NO$_2$@F127 and its applications.** The fabrication of the nanoprobe BTPE-NO$_2$@F127 and the biomarker (H$_2$O$_2$)-activatable detection/imaging for the interstitial cystitis, the trazodone-induced liver injury and the liver I/R injury in mouse models.

diseases through the NIR-II fluorescent imaging and the multi-spectral optoacoustic tomography (MSOT) imaging. In addition, the three-dimensional (3D) MSOT images can be attained by employing this nanoprobe, providing 3D information for accurately pinpointing the disease foci.

The molecular probe BTPE-NO$_2$ and the activated probe BTPE-NH$_2$ were synthesized in accordance with the procedures outlined in Supplementary Fig. 1, and resultant products and intermediates were characterized by [1]H NMR, [13]C NMR and mass spectrometry (Supplementary Figs. 2–18). The BTPE-NO$_2$ was then encapsulated with Pluronic F127 to afford the nanop-robe BTPE-NO$_2$@F127. To verify the AIE feature of BTPE-NH$_2$, its fluorescence intensities in the mixture of dimethyl sulfoxide (DMSO)/water (good solvent: DMSO, poor solvent: water) with varying water fraction were measured. Changing the water fraction can fine-tune the solvent mixture's dissolving ability and correspondingly the extent of BTPE-NH$_2$ aggregation. As shown in Fig. 2a, b, as the water fraction increases, the emission intensity of BTPE-NH$_2$ solution increases obviously, exhibiting a typical AIE feature. As the water fraction reaches 99%, the fluorescence intensity for BTPE-NH$_2$ shows about 12.8-fold enhancement as compared to that in DMSO solution.

Owing to the strong hydrophobicity and high conjugation of the probe BTPE-NO$_2$, the high encapsulation efficiency (98%) can be achieved upon being encapsulated by Pluronic F127, and the loading capacity of BTPE-NO$_2$ in the nanoprobe BTPE-NO$_2$@F127 is determined as 9.3%. Transmission electron microscopy (TEM) reveals that the nanoprobe BTPE-NO$_2$@F127 has relatively uniform spherical morphology with a diameter of about 37 nm (inset of Fig. 2c), and the particle size distribution of the nanoprobe BTPE-NO$_2$@F127 measured by the dynamic light scattering (DLS) is shown in Fig. 2c. There are no obvious changes in particle size for the nanoprobe BTPE-NO$_2$@F127 in PBS (pH 7.4) or in PBS containing 10% FBS upon being stored for 7 days (Supplementary Fig. 19a), indicating that the nanoprobe BTPE-NO$_2$@F127 exhibits good storage stability. Additionally, an obvious Tyndall phenomenon can be observed for the nanoparticle dispersion before and after incubation with H$_2$O$_2$ under laser beam irradiation (inset of Supplementary Fig. 19b).

**Optical and acoustic response of BTPE-NO$_2$@F127 toward H$_2$O$_2$.** The concentration-dependent fluorescence and absorption

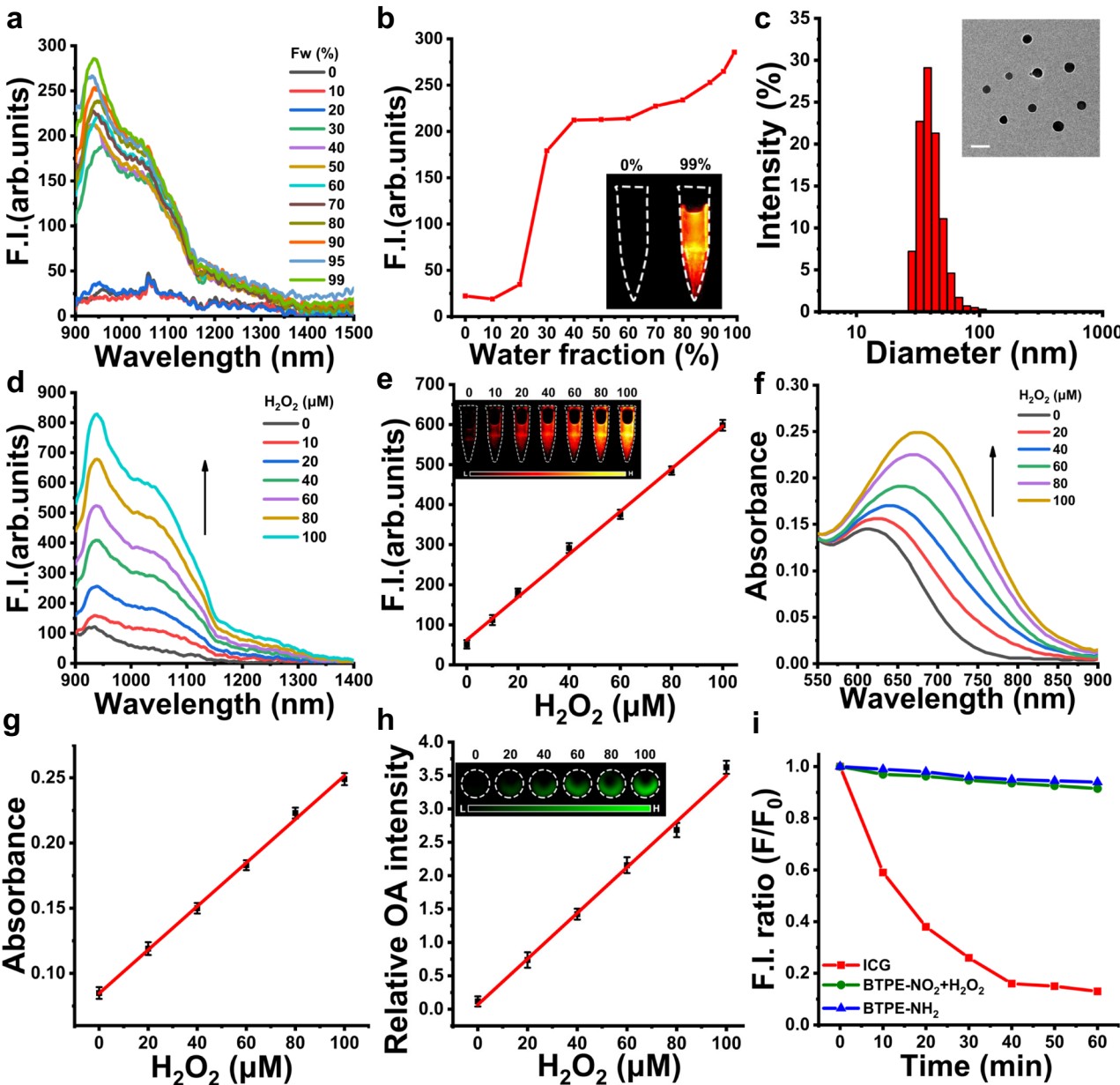

**Fig. 2 Characterization and spectral properties of nanoprobe. a** NIR-II fluorescence spectra of BTPE-NH$_2$ (50 μM) in DMSO/H$_2$O mixtures with different water fractions. Excitation wavelength: 808 nm. **b** Plot of fluorescence intensity (at 938 nm) of BTPE-NH$_2$ versus water fraction. The inset shows NIR-II fluorescence images of BTPE-NH$_2$ in DMSO/H$_2$O with the water fraction being 0% or 99%. **c** Particle size distribution by DLS and TEM images for the nanoprobe BTPE-NO$_2$@F127 (the experiments were repeated independently three times). Scale bar: 50 nm. **d** NIR-II fluorescence spectra of BTPE-NO$_2$@F127 after incubation with varied H$_2$O$_2$ levels (in pH 7.4 PBS) for 90 min. Excitation wavelength: 808 nm. **e** Fluorescence intensity at 1028 nm for the nanoprobe BTPE-NO$_2$@F127 (BTPE-NO$_2$ 32.6 μg mL$^{-1}$) as a function of H$_2$O$_2$ level ($n = 3$ independent experiments). Inset: NIR-II fluorescent images for BTPE-NO$_2$@F127 in the presence of varied levels of H$_2$O$_2$ in PBS; color bar: L: 6.0 × 10$^2$, H: 6.0 × 10$^4$ (arb. units). **f** Absorption spectra for BTPE-NO$_2$@F127 after treatment with varied levels of H$_2$O$_2$ (in PBS). **g** Absorbance at 680 nm for the nanoprobe BTPE-NO$_2$@F127 (BTPE-NO$_2$ 32.6 μg mL$^{-1}$) versus H$_2$O$_2$ level ($n = 3$ independent experiments). **h** Relative optoacoustic intensity for the nanoprobe BTPE-NO$_2$@F127 after incubation with varied levels of H$_2$O$_2$ ($n = 3$ independent experiments, excitation wavelength: 680 nm). Inset: Optoacoustic images for BTPE-NO$_2$@F127 in the presence of varied levels of H$_2$O$_2$ in PBS; color bar: L: 6.1 × 10$^1$, H: 4.1 × 10$^3$ (arb. units). **i** Photostability of BTPE-NH$_2$, BTPE-NO$_2$ upon incubation with 100 μM H$_2$O$_2$, and ICG under the continuous irradiation by 808 nm laser (80 m W cm$^{-2}$) for 60 min. F.I. fluorescence intensity, OA optoacoustic. Data with error bars are all presented as mean ± SD.

spectra for BTPE-NO$_2$@F127 after treatment with H$_2$O$_2$ of different concentrations were measured in PBS (10 mM, pH = 7.4) under 37 °C (Fig. 2d–g). The fluorescence intensity at 1028 nm increases with increasing H$_2$O$_2$ levels (Fig. 2d–g), while the nanoprobe BTPE-NO$_2$@F127 is weakly emissive in the absence of H$_2$O$_2$. The absorption spectra for the nanoprobe BTPE-NO$_2$@F127 before or after H$_2$O$_2$ incubation are presented in

Fig. 2f. Before responding to H$_2$O$_2$, the nanoprobe BTPE-NO$_2$@F127 exhibits the maximum absorption at about 615 nm. After responding to H$_2$O$_2$, the maximum absorption red-shifts to about 680 nm and the absorption band ranges from 600–850 nm. It is obvious from Fig. 2f, g that, with increasing H$_2$O$_2$ concentration, the absorption spectra exhibit a red-shift and the absorbance at 680–850 nm increases evidently. As shown in

Supplementary Fig. 20a, b, the fluorescence spectra of the nanoprobe BTPE-NO$_2$@F127 after incubation with H$_2$O$_2$ show an emission peak at 938 nm and a shoulder peak at 1028 nm, while the absorption peak exhibits a 65 nm red-shift from 615 nm to 680 nm. Additionally, the fluorescence spectra of BTPE-NH$_2$@F127 and BTPE-NO$_2$@F127 were also measured using 615 nm (the maximum absorption of BTPE-NO$_2$@F127) or 808 nm as the excitation light (Supplementary Fig. 20c, d). These data clearly indicate that the increased fluorescence intensity is the result of the probe's response to H$_2$O$_2$. The stability of BTPE-NO$_2$@F127 after responding to H$_2$O$_2$ was investigated as well. It can be clearly seen from Supplementary Fig. 20e that the size and zeta potential of nanoparticles display almost no change for 72 h after responding to H$_2$O$_2$. Furthermore, the fluorescent intensity at 1028 nm remains almost unchanged for 72 h after responding to H$_2$O$_2$ (Supplementary Fig. 20f). The data signify that the BTPE-NO$_2$@F127 probe after responding to H$_2$O$_2$ has a good stability. The time-dependent fluorescence and absorption spectra of BTPE-NO$_2$@F127 nanoprobe after treatment with 100 µM H$_2$O$_2$ for varied time periods are displayed in Supplementary Fig. 21, showing that the reaction between the nanoprobe BTPE-NO$_2$@F127 and H$_2$O$_2$ is complete in about 60 min. The results indicate that BTPE-NO$_2$@F127 nanoprobe is suitable for NIR-II fluorescence imaging.

The optoacoustic intensities in the range of 680–900 nm for the BTPE-NO$_2$@F127 nanoprobe upon the incubation with different concentrations of H$_2$O$_2$ are presented in Fig. 2h and Supplementary Fig. 22a. The linear relationship between the optoacoustic intensities and H$_2$O$_2$ levels (0–100 µM) is clearly observable (Fig. 2h). In addition, the nanoprobe BTPE-NO$_2$@F127 upon incubation with H$_2$O$_2$ and its activable form BTPE-NH$_2$ show much better photostability compared to indocyanine green (ICG) that is FDA-approved for clinical application under the continuous irradiation with 808 nm laser for 60 min (Fig. 2i). To study the anti-interference properties and selectivity of the BTPE-NO$_2$@F127 nanoprobe toward H$_2$O$_2$, a variety of potential interfering substances, such as the biomolecules and ions that are biologically relevant, were tested. It can be seen that the nanoprobe BTPE-NO$_2$@F127 displays good selectivity and anti-interference properties toward H$_2$O$_2$ (Supplementary Fig. 23). The detection limit of BTPE-NO$_2$@F127 toward H$_2$O$_2$ was determined as 0.74 µM (3σ/slope, Supplementary Fig. 22b). Moreover, the maximum absorption of BTPE-NH$_2$ slightly blue-shifts in the low-polar solvents such as toluene and dichloromethane (Supplementary Fig. 22c). BTPE-NH$_2$ is non-emissive (fluorescence quenching) in high-polar solvents such as dimethylformamide and dimethyl sulfoxide, while becoming emissive in the low-polar solvents (toluene or dichloromethane, Supplementary Fig. 22d). The molar extinction coefficient of BTPE-NO$_2$ is $0.58 \times 10^4$ L·mol$^{-1}$·cm$^{-1}$ at 615 nm and that of BTPE-NH$_2$ is $1.07 \times 10^4$ L·mol$^{-1}$·cm$^{-1}$ at 680 nm in pH 7.4 PBS containing 5% DMSO. The mass extinction coefficient of BTPE-NO$_2$@F127 is 4.45 mL·mg$^{-1}$·cm$^{-1}$ at 615 nm and BTPE-NH$_2$@F127 is 10.32 mL·mg$^{-1}$·cm$^{-1}$ at 680 nm in pH 7.4 PBS.

It is posited that upon reaction with H$_2$O$_2$, the probe compound BTPE-NO$_2$ is converted into the chromophore BTPE-NH$_2$ (the activated form of the probe). To verify this response mechanism, $^1$H NMR spectra were measured for the probe BTPE-NO$_2$ before and after its reaction with H$_2$O$_2$. The probe BTPE-NO$_2$ before reaction with H$_2$O$_2$ shows the proton peaks at 8.99 ppm, 8.58–8.59 ppm and 8.33–8.34 ppm (Supplementary Fig. 24), corresponding to the protons of amide group and the protons on the ortho-position and the meta-position of nitrophenyl respectively. After complete reaction of the probe BTPE-NO$_2$ with H$_2$O$_2$, the above-mentioned proton peaks disappear as a result of the product BTPE-NH$_2$ being formed.

Moreover, mass spectrum was measured for the probe BTPE-NO$_2$ after its incomplete reaction with H$_2$O$_2$. The peak at m/z = 1577.4082 [M + H]$^+$ and the peak at m/z = 1223.3958 [M + H]$^+$ correspond to the probe BTPE-NO$_2$ and the chromophore BTPE-NH$_2$, respectively (Supplementary Fig. 25a). In addition, the absorption and emission spectra for BTPE-NH$_2$@F127 and BTPE-NO$_2$@F127 upon reaction with H$_2$O$_2$ were measured (Supplementary Fig. 25b, c). It is clearly apparent that their spectra are quite similar. These results signify that upon reaction with H$_2$O$_2$, the probe BTPE-NO$_2$ is transformed into the chromophore BTPE-NH$_2$ (the activated form of the probe). In addition, density functional theory (DFT) calculations were performed for the electronic structures of BTPE-NO$_2$ and BTPE-NH$_2$. As shown in Supplementary Fig. 26, the highest occupied molecular orbitals (HOMO) of both BTPE-NO$_2$ and BTPE-NH$_2$ are delocalized along the conjugated backbone, and the lowest unoccupied molecular orbital (LUMO) of the BTPE-NO$_2$ is located mainly on the nitrophenyloxoacetamide units, while the LUMO of BTPE-NH$_2$ is located mainly on the central benzothiadiazole core. The energy gap between the HOMO and LUMO in BTPE-NH$_2$ is 1.85 eV, which is lower than that for BTPE-NO$_2$ (2.01 eV). The DFT calculations also confirm the spectral difference between BTPE-NO$_2$ and BTPE-NH$_2$.

**Cell experiments and biosafety of BTPE-NO$_2$@F127.** The ability of the nanoprobe BTPE-NO$_2$@F127 to respond to H$_2$O$_2$ in living cells was evaluated by cell imaging studies. First, the cytotoxicity of the nanoprobe BTPE-NO$_2$@F127 was evaluated using RAW264.7 cells by MTT (3-(4,5-dimethylthiazol-2-yl)-2,5-diphenyltetrazolium bromide) assay. As shown in Supplementary Fig. 27, the nanoprobe BTPE-NO$_2$@F127 exhibits low cytotoxicity toward the cell line. Notably, even after being incubated with the nanoprobe BTPE-NO$_2$@F127 at concentration of 500 µg mL$^{-1}$ (equivalent to 46.5 µg mL$^{-1}$ BTPE-NO$_2$), the viabilities of the cells are still higher than 85%. Hence, the low toxic nanoprobe BTPE-NO$_2$@F127 is suitable for cell imaging. Afterwards, the cell imaging was performed upon RAW264.7 cells being incubated with different concentrations of H$_2$O$_2$ (Supplementary Fig. 28). The NIR-II fluorescence intensities and optoacoustic intensities enhance upon the exposure of RAW264.7 cells to higher H$_2$O$_2$ concentrations (from 0 to 200 µM). The above results suggest that the nanoprobe BTPE-NO$_2$@F127 is promising for detecting the biomarker H$_2$O$_2$ with optoacoustic imaging and NIR-II fluorescence imaging.

The biosafety of the nanoprobe BTPE-NO$_2$@F127 was investigated by histological analysis for the major organs and the body weight change monitoring of the healthy mice intravenously (i.v.) injected with the nanoprobe BTPE-NO$_2$@F127 and saline (as the control) respectively. As shown in Supplementary Figs. 29 and 30a, it is evident that there are no apparent histopathological morphology aberrancies in the organ tissues from the control and the group injected with BTPE-NO$_2$@F127, and there are insignificant differences in the body weights for the control group and the group injected with BTPE-NO$_2$@F127. The observations denote that BTPE-NO$_2$@F127 nanoprobe exhibits high biosafety. Moreover, the aspartate aminotransferase (AST) and alanine aminotransferase (ALT) levels show no abnormality in the group of healthy mice with post nanoprobe injection for 7 days compared to the control (healthy mice injected with saline, Supplementary Fig. 30b). To evaluate the biodistribution, pharmacokinetics, and long-term clearance of the nanoparticles, BTPE-NH$_2$@F127 was employed in the healthy mice. The nanoparticles are first observable in bloodstream and then cleared from the bloodstream with accumulation in the liver within 4 h (Supplementary Fig. 30c). Furthermore, the nanoparticles are distributed mainly in the liver

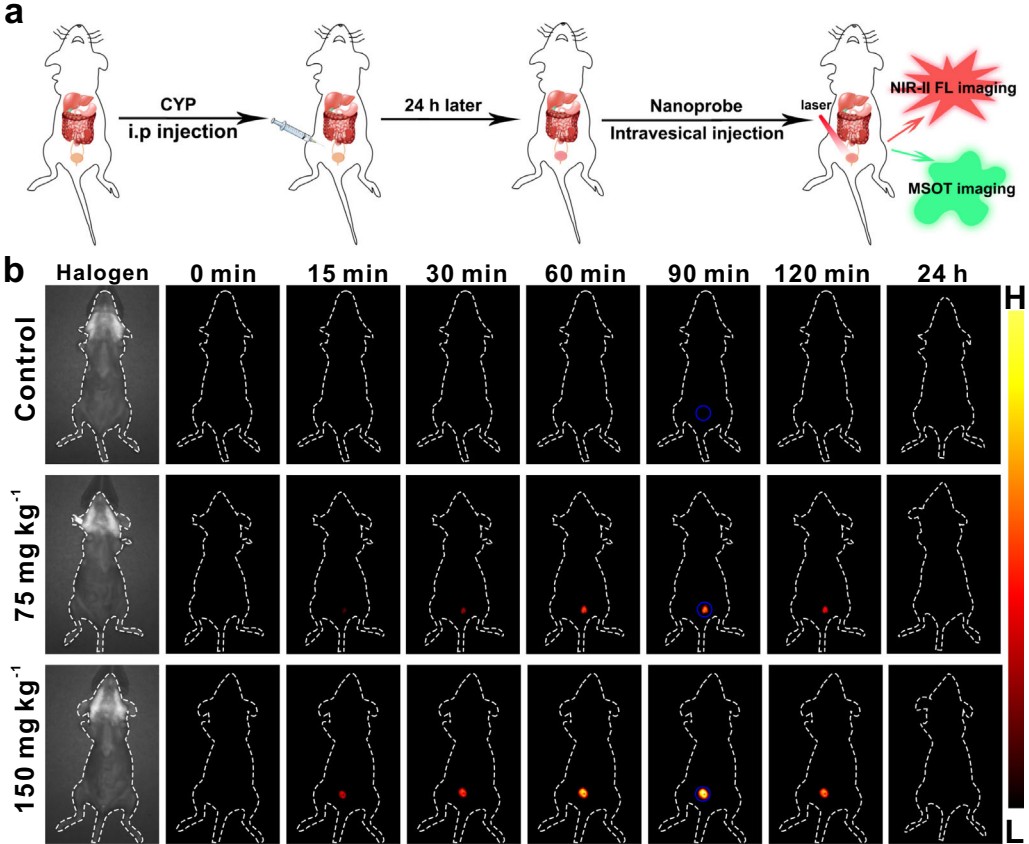

**Fig. 3 Application of nanoprobe BTPE-NO$_2$@F127 in interstitial cystitis mouse model via NIR-II fluorescence imaging. a** Schematic illustration for establishment of the interstitial cystitis mouse model and imaging experiment. **b** Representative NIR-II fluorescence images of the control (healthy mice intraperitoneally injected with saline) and the interstitial cystitis model groups (intraperitoneally injected with 75 mg kg$^{-1}$ CYP or 150 mg kg$^{-1}$ CYP for 24 h) at various time points post-injection of BTPE-NO$_2$@F127. Color bar: L: $6.0 \times 10^2$, H: $6.0 \times 10^4$ (arb. units). The mice were in supine position. Blue circle: bladder region. CYP cyclophosphamide, FL fluorescence, MSOT multispectral optoacoustic tomography.

and eventually cleared out of the body via the intestine. Most of BTPE-NH$_2$@F127 are cleared out post injection for 48 h (Supplementary Fig. 30d).

**Imaging of interstitial cystitis mouse model by BTPE-NO$_2$@F127.** Based on the above encouraging results confirming that the nanoprobe BTPE-NO$_2$@F127 can effectively respond to H$_2$O$_2$ and thus generate NIR-II fluorescent and optoacoustic signals, we then applied the nanoprobe for diagnosing several inflammatory diseases. Interstitial cystitis is an inflammatory condition of bladder featured with pelvic pain as well as urine frequency and urgency. It is hard to diagnose, as currently no specific clinical testing exists to diagnose the disease and hence it is often diagnosed upon other conditions being ruled out[45]. There exists the enhanced production of ROS including H$_2$O$_2$ in the bladder in the case of interstitial cystitis, hence H$_2$O$_2$ can serve as an in situ biomarker for the disease[46,47]. We thus employed the nanoprobe BTPE-NO$_2$@F127 to diagnose interstitial cystitis via responding to the in situ biomarker H$_2$O$_2$. Cyclophosphamide (CYP, trade name Cytoxan or Neosar) is an FDA-approved chemotherapy medication, and inducing interstitial cystitis is one of its well-known adverse effects[48,49]. It is known that CYP causes interstitial cystitis through the following process: once CYP enters the body and is metabolized in the liver, an active antitumor drug and a toxic metabolite (acrolein) are produced, and acrolein is excreted in the urine through kidney, bladder and urethra, during this process acrolein induces bladder damage by injuring the bladder's transitional epithelium[50]. Therefore, we used CYP to

establish the interstitial cystitis mouse model by intraperitoneal injection of varied doses of CYP. As shown in Fig. 3a, when CYP was intraperitoneally administered to female mice for 24 h, the nanoprobe BTPE-NO$_2$@F127 (BTPE-NO$_2$: 2.3 mg kg$^{-1}$) was intravesically injected into the mice, and then the NIR-II fluorescence imaging and MSOT imaging were performed. From the NIR-II fluorescent imaging results (Fig. 3b and Supplementary Fig. 31a), the fluorescence intensities increase gradually with time and reach the maximum in about 90 min post injection of BTPE-NO$_2$@F127, and then gradually fade due to the metabolism. The overexpressed H$_2$O$_2$ in the bladder activates the nanoprobe and thus gives out the NIR-II fluorescent signals. At 90 min post BTPE-NO$_2$@F127 injection, the group of mice with high CYP dose (150 mg kg$^{-1}$) exhibits higher fluorescence intensity compared to the low CYP dose (75 mg kg$^{-1}$) group and the control group (healthy mice), suggesting that higher dose of CYP causes more severe injuries in bladder region.

Unlike fluorescent imaging that generates proton signals, the optoacoustic imaging produces ultrasound signals as the reporting signals upon excitation, and hence it avoids strong proton scattering by biological tissues and offers high penetration depth with high spatiotemporal resolution[37–43]. For MSOT imaging, the biological samples are irradiated by lasers with multiple wavelengths, and the ultrasound signals produced by various photoabsorbers (endogenous photoabsorbers including hemoglobin and exogenous probes) in the samples are detected and collected[51–55]. Then, spectral un-mixing based on specific absorption (or optoacoustic) spectrum of each photoabsorber is

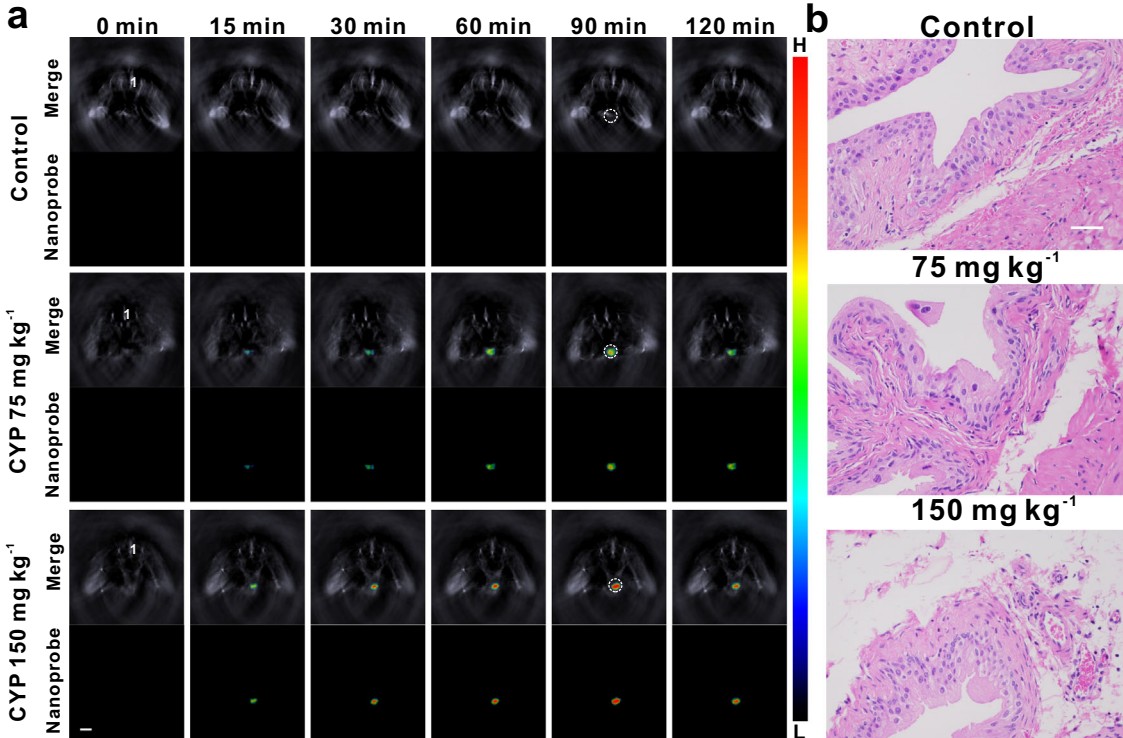

**Fig. 4 Application of nanoprobe BTPE-NO$_2$@F127 in interstitial cystitis mouse model via MSOT imaging and H&E analysis. a** Representative cross-sectional MSOT images of the control mice (the healthy mice intraperitoneally injected with saline) and the interstitial cystitis model groups (24 h after intraperitoneal injection with 75 mg kg$^{-1}$ CYP or 150 mg kg$^{-1}$ CYP) at various time points post intravesical injection of BTPE-NO$_2$@F127. The mice were in prone position. Upper panel: overlay of the activated probe signal with the background (grayscale) signal. Lower panel: multispectrally resolved signal from the activated probe. Organ labeling: 1 artery. White dotted circle: bladder region. Color bar: L: $6.1 \times 10^1$, H: $4.1 \times 10^3$ (arb. units). **b** Tissue sections (H&E staining) for bladders of various mice groups ($n = 5$ animals per group. The experiments were repeated independently three times with similar results). Scale bar: 50 μm. CYP cyclophosphamide.

conducted to separately visualize signals from individual photo-absorber. In addition to its advantages including high spatial resolution and large penetration depth for imaging, MSOT imaging can generate 3D images after the cross-sectional (tomographic) images being stacked and rendered as the 3D maximum intensity projection images, enabling accurate location of the site of interest (e.g., the disease site) non-invasively and in a spatiotemporal way. The MSOT imaging with the aid of the nanoprobe BTPE-NO$_2$@F127 (BTPE-NO$_2$: 2.3 mg kg$^{-1}$; the nanoprobe was administered via intravesical injection) was performed for interstitial cystitis mouse model, and the results are shown in Fig. 4a and Supplementary Fig. 31. A female mouse cryosection image that corresponds to the bladder cross-section is shown in Supplementary Fig. 31b, and the mean MSOT intensities for the bladder region in the cross-sectional images are presented in Supplementary Fig. 31c. By comparing the cross-sectional MSOT image to a cryosection image (Fig. 4a and Supplementary Fig. 31b), it is apparently plain that the MSOT signals are indeed at bladder site, and the signals reach the maximum at about 90 min upon intravesical injection of the nanoprobe BTPE-NO$_2$@F127. The 3D orthogonal-view MSOT images (3D MIP images) are obtained to locate the bladder inflammation, and the 3D MSOT images for the control (healthy mice treated with saline) and the model mice groups upon treatment with varied doses of CYP are exhibited in Supplementary Fig. 31d. With the 3D MSOT images, the location of the bladder inflammation can be clearly observed. In addition, histological analyses (with H&E staining) were conducted for the bladder tissues from the control and the interstitial cystitis model mice (Fig. 4b). It is lucid that for the model groups treated

with CYP, hemorrhage, edema and urothelial sloughing can be observed, and the group with higher CYP dose suffers more severe conditions. These results confirm that the nanoprobe BTPE-NO$_2$@F127 can detect the inflamed bladder via responding to the in situ biomarker H$_2$O$_2$ and thus diagnose the interstitial cystitis non-invasively via the NIR-II fluorescent and MSOT imaging, and the 3D MSOT image could identify and locate the disease site.

**Imaging of liver injury mouse model by BTPE-NO$_2$@F127.** Trazodone is a clinical medicine for treating depression, and high daily dose of trazodone would cause liver injury. It is well known that ROS is overexpressed in hepatic region in case of liver injury, and thus hepatic H$_2$O$_2$ acts as the endogenous biomarker for liver damage or injury[56,57]. Trazodone (with its structure shown in Fig. 5a) was employed to stimulate hepatic injury in male mice by intraperitoneal injection of varied doses of trazodone for three consecutive days. 6 h after the last injection of trazodone (on the third day), the serum levels of the enzyme ALT from different groups were measured by Elisa kits. It was observed that the serum ALT levels of the mice increase with the increasing trazodone dosage (Fig. 5b), confirming the increasing severity of liver injury with increasing trazodone dose. Next, we employed the nanoprobe BTPE-NO$_2$@F127 to image the liver injuries. The NIR-II fluorescence images for the control group and the groups administered with different amounts of trazodone at 0 min and 120 min upon intravenous (i.v.) injection of BTPE-NO$_2$@F127 are shown Fig. 5c and Supplementary Fig. 32. For the mice treated with trazodone, no fluorescence can be observed at 0 min before injection of the nanoprobe BTPE-NO$_2$@F127, while the

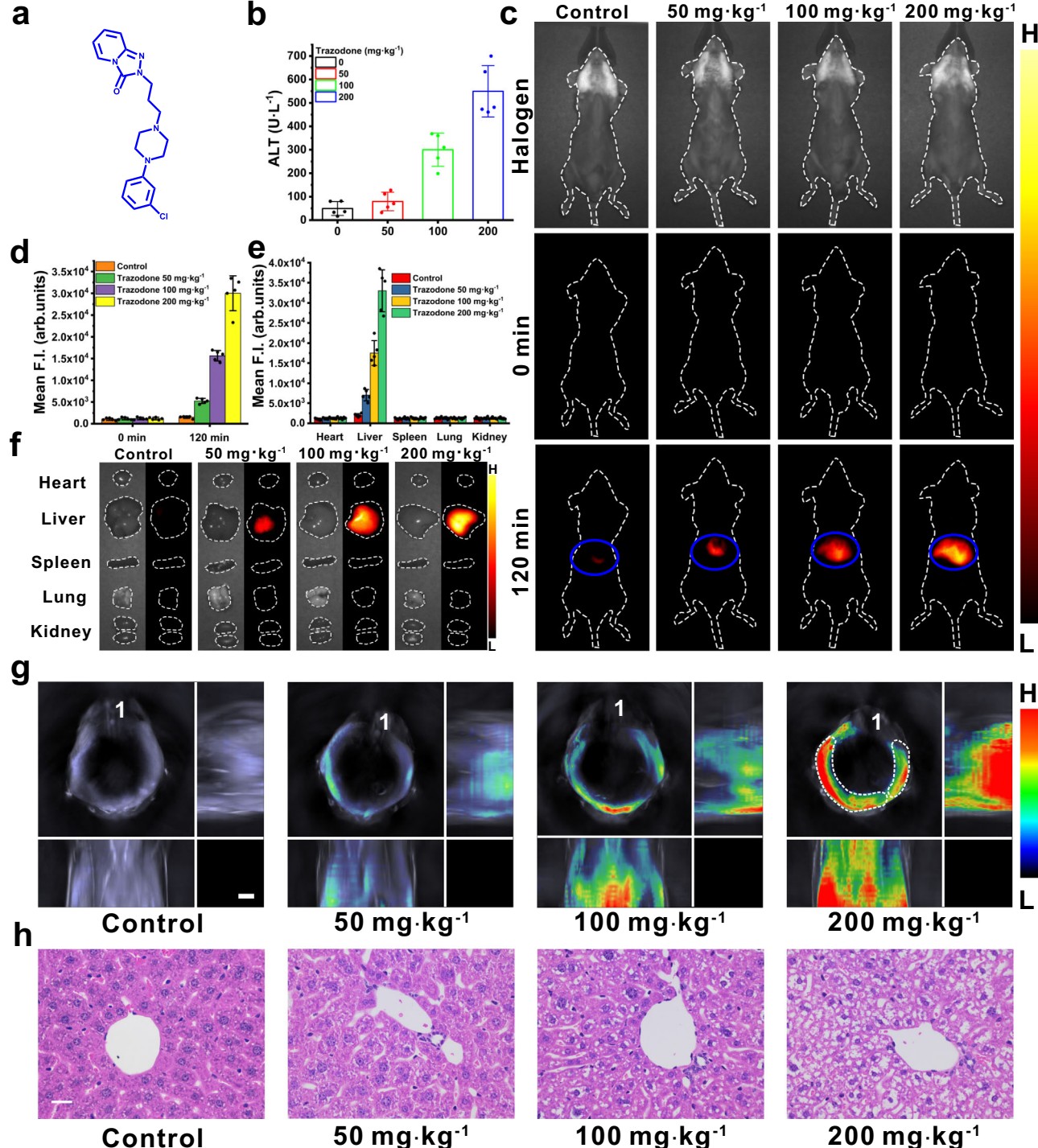

**Fig. 5 Application of nanoprobe BTPE-NO$_2$@F127 in trazodone-induced liver injury mouse model. a** Molecular structure of trazodone. **b** Serum levels for the enzyme ALT of different groups ($n = 5$ animals per group). **c** Typical photographs under halogen light (serving as bright-field images) and NIR-II fluorescence images for the control group (mice administered with saline), and the mice pretreated with 50 mg kg$^{-1}$, 100 mg kg$^{-1}$ and 200 mg kg$^{-1}$ trazodone at 0 min or at 120 min after i.v. injection of the nanoprobe BTPE-NO$_2$@F127. Blue circle: liver region. Color bar: L: $6.0 \times 10^2$, H: $6.0 \times 10^4$ (arb. units). **d** Mean NIR-II fluorescence intensities at ROI (circled with blue curve) for liver site of the mice in **c**. $n = 5$ animals per group. **e** Mean NIR-II fluorescence intensities of the major organs corresponding to NIR-II fluorescent images in **f**. $n = 5$ animals per group. **f** Representative ex vivo NIR-II fluorescent images for the dissected organs (heart, liver, spleen, lung, and kidney) from different groups at 120 min after i.v. injection of BTPE-NO$_2$@F127. Color bar: L: $6.0 \times 10^2$, H: $6.0 \times 10^4$ (arb. units). **g** Typical 3D MSOT images for the mice at 120 min after BTPE-NO$_2$@F127 injection. White dotted circle: liver region. Scale bar: 3 mm. **h** H&E stained liver sections of different groups ($n = 5$ animals per group. The experiments were repeated independently three times with similar results). Color bar: L: $6.1 \times 10^1$, H: $4.1 \times 10^3$ (arb. units). Scale bar: 20 μm. Data with error bars are all presented as mean ± SD. F.I. fluorescence intensity, ALT alanine aminotransferase.

fluorescent intensities increase with the increasing dose of trazodone at 120 min after the injection of the nanoprobe BTPE-$NO_2$@F127. The quantified intensities of the region of interest (ROI) covering the liver region (Fig. 5c and Supplementary Fig. 32) are presented in Fig. 5d and Supplementary Fig. 33, which provide intuitive data for fluorescence images. The fluorescent intensities of the excised organs (kidney, heart, liver, lung and spleen) from different groups of mice at 120 min after the treatment of the nanoprobe BTPE-$NO_2$@F127 are shown in Fig. 5e, and the NIR-II fluorescent images of these organs are revealed in Fig. 5f. The liver exhibits much stronger fluorescence than other major organs such as heart, spleen, lung and kidney, and the fluorescence intensities from the groups treated with different doses of trazodone show the dose-dependent behavior.

Afterwards, the nanoprobe BTPE-$NO_2$@F127 was utilized to observe the liver injury stimulated by trazodone via MSOT imaging. Supplementary Fig. 34 presents cross-sectional images of healthy mice that were pretreated with saline (control) and the mice that were pretreated with different doses of trazodone before (0 min) and after i.v. injection of the BTPE-$NO_2$@F127 nanoprobe for varied times. The upper panel of the figure displays the overlay of the image from activated nanoprobe and that from 850 nm single-wavelength excitation (namely, the grayscale background serving as an anatomical reference), and the lower panel provides the color images showing the biodistribution of the activated nanoprobe's signals via spectral unmixing. The signals of activated nanoprobe are apparently observable and progressively enhance with time after the injection of the BTPE-$NO_2$@F127 nanoprobe, and their intensities attain the peak level at 120 min. The mean signal intensity for the liver region (ROI) in different groups of mice is given in Supplementary Fig. 35. Moreover, the 3D orthogonal-view MSOT images covering the liver region of different groups of mice are presented in Fig. 5g, which are capable of reflecting 3D information on liver injury of different groups of mice. It can be seen that weak MSOT signals appear in the hepatic area of the mice without trazodone treatment after nanoprobe injection, because low ROS (including $H_2O_2$) level exists in the liver region in healthy mice[58]. For the groups subject to treatment of varied amounts of trazodone (50, 100 or 200 mg kg$^{-1}$), prominent MSOT signals are found at their liver regions, and the signals are stronger for the mice treated with higher trazodone dose, indicating the mice with higher dose of trazodone suffer more severe liver damage. These results signify that the increase in $H_2O_2$ level in the hepatic region resulted from liver injury can be detected using MSOT imaging in a spatiotemporal way. Afterwards, the mice pretreated with 0, 50, 100 and 200 mg kg$^{-1}$ of trazodone were euthanized, from which the main organs were acquired and underwent ex vivo imaging. As revealed by Supplementary Figs. 36 and 37, the livers display strongest MSOT signals among these main organs. Next, histological analysis (H&E staining) for the tissue sections of livers from different groups of mice was conducted as well (Fig. 5h). The liver tissue section from the control group shows normal morphology with no obvious aberration, while those from the mice treated with trazodone display nuclear cytoplasmic separation, vacuolization and hydropic degeneration, and high dose of trazodone enhances the liver injury severity.

Furthermore, we used the nanoprobe to image another inflammatory disease. Liver ischemia-reperfusion (I/R) injury is a major complication in a variety of clinical scenarios, such as liver resection, liver transplantation, hemorrhagic shock, resuscitation and trauma surgery, and it remains the main cause of mortality or morbidity because of graft rejection upon liver transplantation[59,60]. In the case of liver I/R injury, ROS (including $H_2O_2$) is overexpressed in the hepatic region, which leads to local inflammatory responses and cell apoptosis and further aggravates

liver injury[61,62]. Herein, we employed the nanoprobe BTPE-$NO_2$@F127 to detect the degree of liver injuries on account of its response to hepatic $H_2O_2$. The mice were treated with ischemia for 0 min (the sham group serving as the control), 30 min or 60 min, followed by reperfusion for 24 h. Figure 6a shows the liver I/R process of the mice. In the case of ischemia, the liver appears pale colored due to the blocked blood supply, while sanguineous perfusion is clearly observable upon the vascular clamp being removed. For the sham group and the I/R model groups, the nanoprobe BTPE-$NO_2$@F127 was i.v. injected into different groups of mice and then the mice underwent MSOT imaging and NIR-II fluorescence imaging. The NIR-II fluorescence images and the fluorescence intensities of the liver region of the mice are shown in Fig. 6b, c. The liver area of the mice in the sham group exhibits only weak fluorescence at 90 min or 120 min upon the injection of the nanoprobe BTPE-$NO_2$@F127, while the liver areas of the ischemia groups show evident fluorescence. The long ischemia group (ischemia for 60 min) exhibits higher fluorescent intensities compared to the short ischemia group (ischemia for 30 min), because higher amount of hepatic $H_2O_2$ is generated for longer ischemia time and correspondingly more severe liver damages ensue for the long ischemia group. For the groups with 30 or 60 min of ischemia in the I/R process (Fig. 6d and Supplementary Fig. 38), the liver presents much stronger fluorescence among the main organs, which is consistent with the in vivo imaging results.

Then, the nanoprobe BTPE-$NO_2$@F127 was used to detect liver I/R injuries via MSOT imaging. The related data are presented in Fig. 7a and Supplementary Fig. 39. Figure 7a presents the cross-sectional MSOT images of the mice from different groups at varied time points after i.v. injection of BTPE-$NO_2$@F127. The quantified data of the MSOT intensity in the ROI for the liver region are presented in Fig. 7b and Supplementary Fig. 40. Serving as the reference for organ locations, the cryosection image of a male mouse that corresponds to mouse liver region is shown in Fig. 7c. The control (sham group) does not display remarkable MSOT signal in the liver region at 120 min after i.v. injection of BTPE-$NO_2$@F127 nanoprobe (Fig. 7a, b), while the mice from the ischemia groups show obvious MSOT signal at 120 min, indicating the formation of liver injuries. As for the long ischemia group (ischemia 60 min), the signal in the liver region becomes much more conspicuous compared to the control (ischemia 0 min) or the short ischemia group (ischemia 30 min). To verify the relationship between I/R process and liver injury, we then measured the enzyme activities of serum ALT and AST by using Elisa kits. The I/R process leads to remarkable enhancement in the activities of these serum enzymes (Fig. 7d), proving that direct correlation between liver dysfunction and I/R process does exist, and the results are similar to literature reports[62,63].

As one of the first volume visualization methods, maximal intensity projection images (MIP images) usually serve as a method for 3D data in orthogonal views. In this study, we acquired the 3D orthogonal-view images. Figure 7e reveals the 3D MSOT images for the different groups of mice with I/R process (namely sham group, shore ischemia group and long ischemia group) at 120 min after i.v. injection of the nanoprobe BTPE-$NO_2$@F127. As for the group undergoing I/R process with longer time of ischemia (60 min), the injured liver can be distinctly observed with strong MSOT signals, and the volume (size) of the injured liver is much larger compared to the group undergoing I/R process with short time of ischemia (30 min). Next, dissected organs from different groups of mice after euthanasia were subjected to MSOT imaging (Supplementary Figs. 41 and 42). For the groups undergoing I/R process with 30 or 60 min of ischemia, the livers display much more evident MSOT signals than other organs, and the result verify the in vivo imaging observations. Moreover, the liver sections of the mice in different groups were

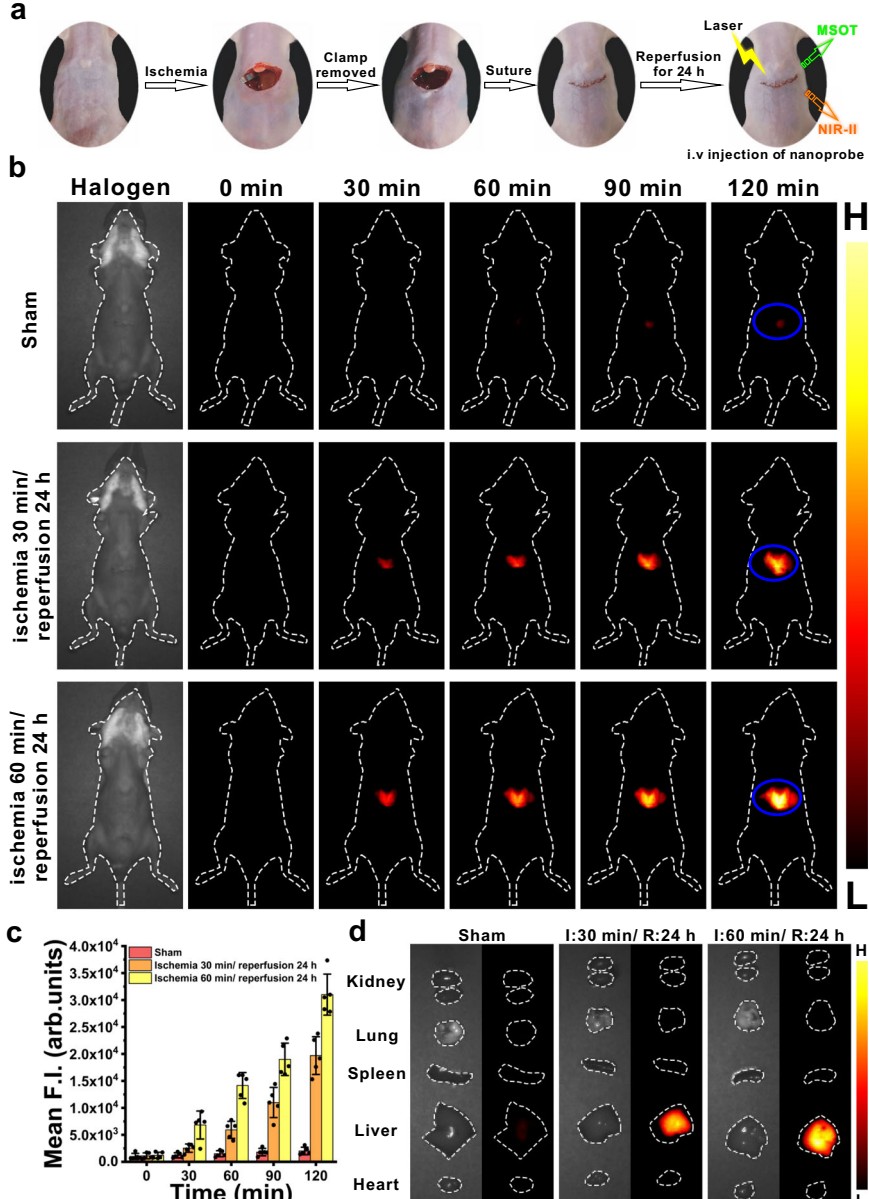

**Fig. 6 Application of nanoprobe BTPE-NO₂@F127 in liver ischemia-reperfusion injury mouse model via NIR-II fluorescence imaging. a** Representative photographs showing the surgery process for the I/R process: the mouse undergoing laparotomy, ischemia with its hepatic artery and portal vein being clamped for 30 or 60 min, the vascular clamp being removed and the suturing procedure. **b** NIR-II fluorescence images for the sham-surgery group (ischemia for 0 min) and the I/R model mice (with ischemia for 30 min or 60 min followed by reperfusion for 24 h) at varied time points after i.v. injection of BTPE-NO₂@F127 nanoprobe. Blue circle: liver region. Color bar: L: 6.0 × 10² , H: 6.0 × 10⁴ (arb. units). **c** Mean NIR-II fluorescence intensities of the liver regions corresponding to the ROI (blue circle) in **b**. $n = 5$ animals per group. Data with error bars are presented as mean ± SD. **d** Representative ex vivo NIR-II images of the dissected major organs (kidney, lung, spleen, heart and liver) in different mouse groups at 120 min after i.v. injection of BTPE-NO₂@F127 nanoprobe. I ischemia, R reperfusion. Color bar: L: 6.0 × 10² , H: 6.0 × 10⁴ (arb. units). MSOT multispectral optoacoustic tomography, NIR-II near-infrared second window, F.I. fluorescence intensity, I ischemia, R reperfusion.

analyzed via histological analyses (Fig. 7f). The inflammatory cell infiltration and vacuolization degeneration are clearly observable in the long ischemia (60 min) group or the short ischemia group (30 min) compared to the sham group, and the conditions are more serious in long ischemia group than those in the short ischemia group, indicating the mice with I/R process suffering severe hepatic injury.

## Discussion

Benzothiadiazole-core chromophores possess promising advantages such as excellent stability, large stoke shift and ease of

extending emission into NIR-II range. However, impediments including poor water solubility and resultant ACQ as well as non-activatable response still need to be circumvented before they can give a full play to their potentials in biological applications. In this study, we have shown that these impediments can be overcome by the design strategy through integrating biomarker-responsive moieties and adopting the self-assembly with encapsulation of the molecular probe.

The comparison between the nanoprobe BTPE-NO₂@F127 with other reported NIR-II fluorescent probes and optoacoustic (photoacoustic) probes, as well as their applications in mouse

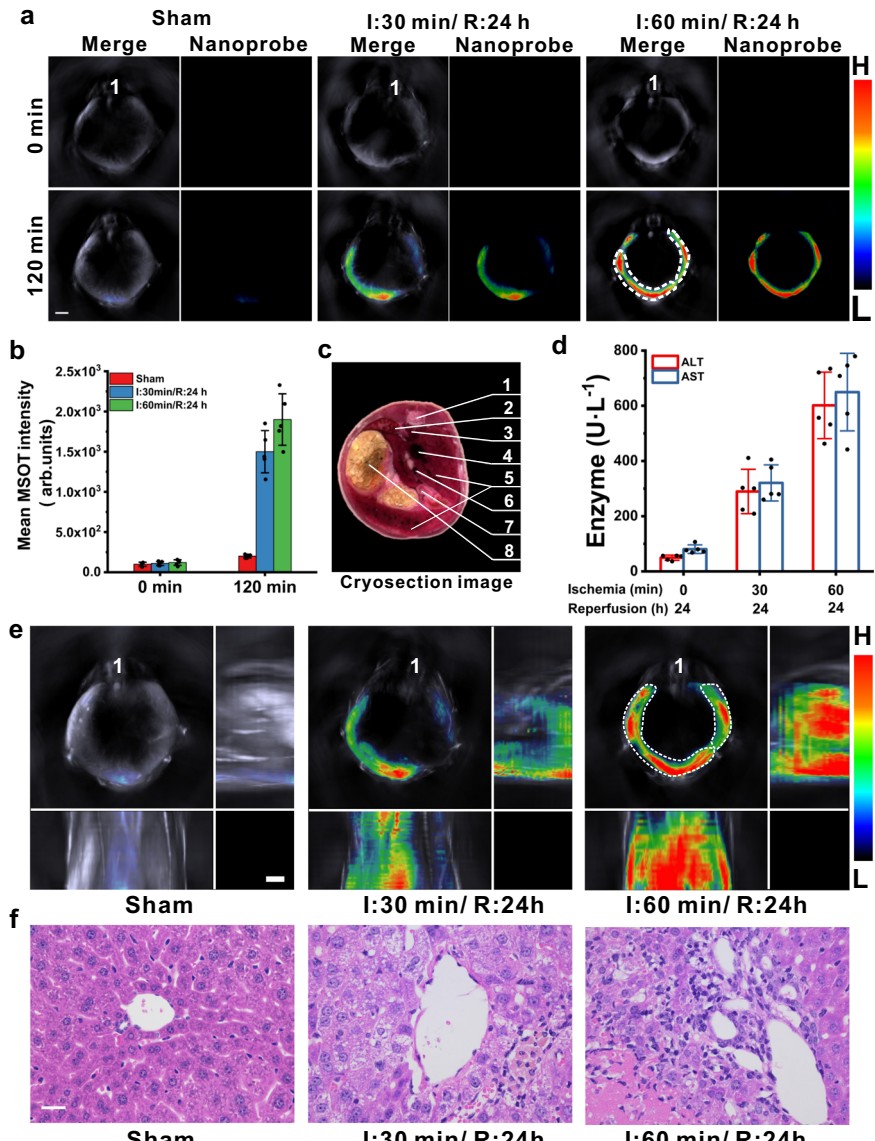

**Fig. 7 Application of nanoprobe BTPE-NO₂@F127 in liver ischemia-reperfusion injury mouse model via MSOT imaging. a** Representative cross-sectional MSOT images for different groups at 0 min and 120 min after i.v. injection of the nanoprobe BTPE-NO₂@F127 (I ischemia, R reperfusion). Organ labeling: "1" represents spinal cord. White dotted circle: liver region. Scale bar: 3.0 mm. Color bar: L: $6.1 \times 10^1$, H: $4.1 \times 10^3$ (arb. units). **b** Mean MSOT intensities corresponding to ROI (white dotted line) in **a**. $n = 5$ animals per group. **c** Cryosection image of a male mouse with its cross-section location matching those in **a**. 1: Spinal cord; 2: Spleen; 3: Thoracic aorta; 4: Vena cava; 5: Liver; 6: Portal vein; 7: Intestines; 8: Stomach. The cryosection image is from CryoMOUSE^TM atlas provided with the MSOT equipment for anatomical reference. **d** Serum levels of two enzymes ALT and AST from various groups. $n = 5$ animals per group. **e** Typical 3D MSOT images for different groups at 120 min post-injection of BTPE-NO₂@F127 nanoprobe. "1" represents spinal cord. White dotted circle: liver region. Color bar: L: $6.1 \times 10^1$, H: $4.1 \times 10^3$ (arb. units). **f** Representative H&E staining of liver sections from different mouse groups ($n = 5$ animals per group. The experiments were repeated independently three times with similar results. Scale bar: 20 μm). Data with error bars are all presented as mean ± SD. I ischemia, R reperfusion, MSOT multispectral optoacoustic tomography.

disease models clearly demonstrates that our nanoprobe outperforms other NIR-II and photoacoustic imaging probes in many ways, particularly in terms of activatable response and signal-to-background ratio (Supplementary Tables 1–3).

In summary, an activatable benzothiadiazole-core nanoprobe BTPE-NO₂@F127 with AIE feature has been developed for in vivo NIR-II fluorescent imaging and MSOT imaging. The presence of pathological levels of H₂O₂ in the disease sites activates the nanoprobe through cleaving the recognition groups and generating the chromophore BTPE-NH₂, thereby producing strong NIR-II fluorescent emission and optoacoustic signals. The nanoprobe is able to detect and image the

trazodone-induced liver injury, the I/R liver injury and the interstitial cystitis in mouse models via responding to the in situ H₂O₂ biomarker. The nanoprobe could therefor serve as a robust tool to detect and image the disease sites with MSOT imaging and NIR-II fluorescence imaging. Moreover, the obtained 3D MSOT images are advantageous for visualizing the disease site with 3D information. The study herein unleashes the promising potential of benzothiadiazole-core molecules for optoacoustic imaging and NIR-II fluorescent imaging of specific diseases, and the strategy could offer insights for designing other large π-conjugated NIR-II chromophores for various biological applications.

## Methods

**Reagents**. 4-Aminophenylboronic acid pinacol ester, 4,7-dibromo-5,6-dinitrobenzo[c] [1,2,5] thiadiazole, 1,2-bis(4-bromophenyl)ethane-1,2-dione, iron powder, bis(pinacolato)diborane, bromotriphenylethylene, N-iodosuccinimide (NIS), 4-nitrophenylglyoxylic acid, oxalyl chloride, Pd(PPh$_3$)$_4$, Pd(PPh$_3$)$_2$Cl$_2$, and tributyl(3-methylthiophen-2-yl)stannane were purchased from Aladdin Reagents. Cyclophosphamide, H$_2$O$_2$, and trazodone hydrochloride were purchased from Sigma-Aldrich. DMEM and PBS (10 mM, pH = 7.4) were purchased from KeyGen Bio-Tech. Elisa kits for alanine aminotransferase (ALT) and aspartate transaminase (AST) were purchased from Shanghai Enzyme-linked Biotechnology Co. Ltd. Acetic acid, DMF, K$_2$CO$_3$, THF and all other solvents were analytical-grade reagents. The water used throughout the experiments was triple-distilled water.

**Apparatus**. $^1$H NMR and $^{13}$C NMR spectra were measured with Bruker Avance 600 MHz NMR Spectrometer. High resolution mass spectrometer (HR-MS) was recorded on a Bruker MAXIS IMPACT mass spectrometer. MALDI-TOF was recorded on a MALDI SYNAPT G2-Si mass spectrometer. Near-infrared II (NIR-II) fluorescence spectra were recorded on NIRQUEST512 spectrometer (excitation: 808 nm laser, emission range: 900–1700 nm). The absorption spectra were collected on Hitachi U-3010 spectrophotometer. The NIR-II fluorescence (in vivo and ex vivo) imaging was performed by using NIR-II in Vivo Imaging System (Series II 808/900–1700, Suzhou NIR-Optics Technologies Co., Ltd.). The particle size and distribution were measured through DLS by using a Malvern Nano-ZS90 analyzer at a fixed angle of 90° at 25 °C. TEM experiments were performed with a JEM-1400 TEM transmission electron microscope (copper mesh with ultra-thin carbon film as the sample carrier). Optoacoustic imaging was performed on an inVision128 multispectral optoacoustic tomographic imaging system (iThera Medical GmbH).

**Synthesis of compound 1**. Bromotriphenylethylene (2.0 g, 5.97 mmol), bis(pinacolato)diborane (3.6 g, 14.32 mmol), Pd(PPh$_3$)$_2$Cl$_2$ (420 mg, 0.60 mmol) and K$_2$CO$_3$ (2 M, 12 mL) aqueous solution were dissolved in 1,4-dioxane (48 mL) under nitrogen atmosphere. Then, the mixture was stirred at 90 °C for 48 h. After cooling down to room temperature, the mixture was extracted with ethyl acetate and washed with water for three times. Finally, the product was dried under vacuum to give a white solid without further purification (2.2 g, yield: 96%). $^1$H NMR (600 MHz, Chloroform-d): δ 7.35–7.33 (m, 2H), 7.30–7.29 (m, 3H), 7.14–7.13 (t, J = 7.4 Hz, 2H), 7.08–7.07 (m, 4H), 7.04–7.03 (m, 2H), 6.96–6.95 (m, 2H), 1.12 (s, 12H). HR-MS ESI (m/z) [M + Na]$^+$ [C$_{26}$H$_{27}$BO$_2$Na]$^+$ calcd. 405.2002, found 405.2001.

**Synthesis of compound 2**. 1,2-Bis(4-bromophenyl)ethane-1,2-dione (500 mg, 1.36 mmol) and compound 1 (1250 mg, 3.27 mmol) were dissolved in K$_2$CO$_3$ (2 M, 6 mL) aqueous solution and THF (24 mL), and Pd(PPh$_3$)$_4$ (160 mg, 0.14 mmol) was added under nitrogen protection. After stirring at 80 °C for 24 h, the mixture was cooled down to room temperature and extracted with ethyl acetate. The crude product was purified with silica gel column chromatography with hexane/dichloromethane (2:1, v/v) as the eluent to obtain the white solid (850 mg, yield: 87%). $^1$H NMR (600 MHz, DMSO-d$_6$): δ 7.66–7.64 (d, J = 8.4, 4H), 7.18–7.12 (m, 22H), 6.99–6.97 (m, 12H). HR-MS ESI (m/z) [M + H]$^+$ [C$_{54}$H$_{39}$O$_2$]$^+$ calcd. 719.2950, found 719.2954.

**Synthesis of compound 3**. Under nitrogen atmosphere, 4,7-dibromo-5,6-dinitrobenzo[c] [1,2,5] thiadiazole (500 mg, 1.30 mmol), tributyl(3-methylthiophen-2-yl) stannane (1100 mg, 2.85 mmol) and Pd(PPh$_3$)$_4$ (150 mg, 0.13 mmol) were dissolved in a dry double neck flask with THF (20 mL). The mixture was refluxed for 24 h. After cooling down to room temperature, the solvent was evaporated and the crude product was purified with silica gel column chromatography with hexane/dichloromethane (1:1, v/v) as the eluent to obtain orange-yellow solid (520 mg, yield: 96%). $^1$H NMR (600 MHz, CDCl$_3$): δ 7.57–7.56 (d, J = 5.1, 2H), 7.07–7.06 (d, J = 5.1, 2H), 2.17 (s, 6H). HR-MS ESI (m/z) [M + H]$^+$ [C$_{16}$H$_{11}$N$_4$O$_4$ S$_3$]$^+$ calcd. 418.9942, found 418.9946.

**Synthesis of compound 4**. Compound 3 (500 mg, 1.19 mmol), NIS (645 mg, 2.87 mmol), chloroform (5 mL) and acetic acid (5 mL) were added in a single-necked flask, and stirred at 40 °C in the darkness for 24 h. After the reaction, the solvent was evaporated under reduced pressure. The crude product was dissolved in DMF, added into ice water, and then filtered to collect the orange-red solid (790 mg, yield: 98%). $^1$H NMR (600 MHz, Chloroform-d): δ 7.21 (s, 2H), 2.13 (s, 6H). HR-MS ESI (m/z) [C$_{16}$H$_9$I$_2$N$_4$O$_4$S$_3$]$^+$ [M + H]$^+$ calcd. 670.7875, found 670.7876.

**Synthesis of compound 5**. Compound 4 (400 mg, 0.60 mmol), iron powder (668 mg, 11.94 mmol) and acetic acid (15 mL) were added into a round bottom flask purged with nitrogen. The mixture was stirred vigorously at 80 °C overnight. After that, it was cooled down to room temperature, extracted with ethyl acetate and neutralized with NaHCO$_3$ solution. The crude product was purified with silica gel column chromatography using hexane/dichloromethane (1:1, v/v) as the eluent to obtain yellow solid (350 mg, yield: 96%). $^1$H NMR (600 MHz, DMSO-d$_6$): δ 7.29 (s, 2H), 5.78 (s, 4H), 1.97 (s, 6H). HR-MS ESI (m/z) [M + H]$^+$ [C$_{16}$H$_{13}$I$_2$N$_4$S$_3$]$^+$ calcd. 610.8392, found 610.8388.

**Synthesis of compound 6**. Compound 5 (400 mg, 0.66 mmol) and compound 2 (450 mg, 0.63 mmol) were dissolved in acetic acid (30 mL). The solution was refluxed under nitrogen atmosphere for 24 h. After that, it was cooled down to room temperature, extracted with ethyl acetate and neutralized with NaHCO$_3$ solution. The crude product was purified by silica gel column chromatography with hexane/dichloromethane (2:1, v/v) as the eluent to obtain brown-red solid (790 mg, yield: 97%). $^1$H NMR (600 MHz, CDCl$_3$): δ 7.35–7.34 (d, J = 8.3, 4H), 7.14–7.06 (m, 28H), 7.03–7.01 (d, J = 7.7, 8H), 2.13 (s, 6H). MALDI TOF-MS ESI (m/z) [M + H]$^+$ [C$_{70}$H$_{47}$I$_2$N$_4$S$_3$]$^+$ calcd. 1293.1052, found 1293.2433.

**Synthesis of BTPE-NH$_2$**. Compound 6 (200 mg, 0.15 mmol) and 4-aminophenylboronic acid pinacol ester (85 mg, 0.39 mmol) were dissolved in K$_2$CO$_3$ (2 M, 3 mL) aqueous solution, and THF (12 mL) and Pd(PPh$_3$)$_4$ (26 mg, 0.02 mmol) were added under nitrogen atmosphere. After stirring for 24 h, the mixture was cooled down to room temperature and extracted with ethyl acetate. The crude product was purified by silica gel column chromatography with hexane/ethyl acetate (1:1, v/v) as the eluent to obtain the green solid (130 mg, yield: 69%). $^1$H NMR (600 MHz, DMSO-d$_6$): δ 7.39–7.38 (d, J = 8.6, 4H), 7.28 (s, 2H), 7.16–7.11 (m, 22H), 7.02–6.99 (m, 8H), 6.98–6.96 (d, J = 8.3, 4H), 6.95–6.94 (d, J = 8.1, 4H), 6.63–6.62 (d, J = 8.6, 4H), 5.40 (s, 4H), 2.10 (s, 6H). MALDI TOF-MS ESI [M + H]$^+$ [C$_{82}$H$_{59}$N$_6$S$_3$]$^+$ calcd. 1223.3963, found 1223.3958.

**Synthesis of BTPE-NO$_2$**. Under nitrogen atmosphere, 4-nitrophenylglyoxylic acid (78 mg, 0.40 mmol) and oxalyl chloride (106 μL, 1.20 mmol) were dissolved in dichloromethane (5 mL) and stirred at 0 °C for 3 min, and then DMF (20 μL) was added and refluxed at 45 °C. After 1 h, the solvent and excess oxalyl chloride were removed by rotary vacuum evaporator. The residue was added with BTPE-NH$_2$ (60 mg, 0.05 mmol), triethylamine (56 μL, 0.4 mmol) and dichloromethane (5 mL), and stirred at room temperature for 1 h. The crude product was purified with hexane/ethyl acetate (2:1, v/v) as eluent by silica gel column chromatography to obtain gray-green solid (35 mg, yield: 44%). $^1$H NMR (600 MHz, DMSO-d$_6$): δ 11.16 (s, 2H), 8.42–8.40 (d, J = 8.4, 4H), 8.35–8.33 (d, J = 8.5, 4H), 7.89–7.88 (d, J = 8.2, 4H), 7.77–7.76 (d, J = 8.3, 4H), 7.57 (s, 2H), 7.29–7.27 (m, 4H), 7.17–7.10 (m, 18H), 7.00–6.97 (m, 12H), 6.94–6.93 (s, 4H), 2.15 (s, 6H). $^{13}$C NMR (151 MHz, Chloroform-d): δ 185.93, 157.70, 153.71, 153.14, 150.99, 145.72, 145.25, 143.57, 143.40, 143.16, 142.02, 140.47, 140.30, 137.63, 137.27, 136.09, 135.54, 132.63, 131.98, 131.51, 131.42, 131.30, 131.20, 129.67, 127.85, 127.78, 126.90, 126.87, 126.75, 126.57, 123.58, 120.39, 16.78. MALDI TOF-MS ESI (m/z) [M + H]$^+$ [C$_{98}$H$_{65}$N$_8$O$_8$S$_3$]$^+$ calcd. 1577.4087, found 1577.4082.

**Optical properties of BTPE-NH$_2$ for evaluating AIE feature**. BTPE-NH$_2$ solution was prepared with DMSO at the concentration of 5 mM, which was used as the BTPE-NH$_2$ stock solution. Then, the NIR-II fluorescence emission spectra were recorded with different ratios of DMSO/water (the final concentration of BTPE-NH$_2$: 50 μM).

**Preparation of the nanoprobe BTPE-NO$_2$@F127**. Briefly, the amphiphilic polymer Pluronic F127 (F127, 20 mg) and BTPE-NO$_2$ (2 mg) were dissolved in chloroform (1 mL), followed by sonication for 5 min at room temperature to form a homogeneous solution. Then, the chloroform was removed by vacuum-rotary evaporation. The mixture was resuspended in water and dialyzed against water (containing 1% DMSO, molecular weight cut-off 2000 Da), and then lyophilized.

The dried mixture was resuspended in PBS (pH 7.4, 10 mM). After sonication for 30 min, the resulting dispersion was filtered by using a syringe-driven filter (0.22 μm pore size), and the nanoprobe BTPE-NO$_2$@F127 dispersion was obtained and used directly for the further experiments.

For determining the loading capacity and encapsulation efficiency, the above dried mixture was resuspended in distilled water. After sonication for 30 min, the dispersion was filtered by using a syringe-driven filter (0.22 μm pore size), and the resultant nanoprobe BTPE-NO$_2$@F127 dispersion was lyophilized. The freeze-dried BTPE-NO$_2$@F127 was dissolved in DMSO and then the absorption spectrum was measured. The loading capacity and encapsulation efficiency of the nanoprobe BTPE-NO$_2$@F127 (nanoparticle) were measured by absorption spectroscopy using pre-established standard calibration curves and calculated as follows:

$$\text{Concentration of BTPE} - \text{NO}_2@\text{F127 in water or PBS} = \frac{weight\ of\ nanoparticles}{volume\ of\ solution} \times 100\% \tag{1}$$

$$\text{Loading capacity \%} = \frac{weight\ of\ BTPE - NO_2\ in\ the\ nanoparticles}{weight\ of\ the\ nanoparticles} \times 100\% \tag{2}$$

$$\text{Encapsulation efficiency\%} = \frac{weight\ of\ BTPE - NO_2\ in\ the\ nanoparticles}{weight\ of\ the\ feeding\ BTPE - NO_2} \times 100\% \tag{3}$$

The loading capacity was determined as 9.3% and the encapsulation efficiency was 98%.

**Determination and calculation of fluorescence quantum yield**. The NIR-II fluorescence quantum yield was determined as follows. The NIR-II fluorescent dye IR-26 was used as the reference. IR-26 solutions dissolved in 1,2-dichloroethane were prepared with different concentrations corresponding to different absorbance values at 808 nm, and then the absorption spectra and NIR-II fluorescent spectra (excited at 808 nm) of the solutions were recorded. The absorption spectra and NIR-II fluorescent spectra were measured for the BTPE-NH$_2$@F127 solution (in PBS) and BTPE-NH$_2$ solution (in PBS containing 5% DMSO). All NIR-II fluorescent spectra were integrated in the 900–1400 NIR-II range. The integrated NIR-II fluorescence intensity (900–1400 nm) was plotted against the absorbance at 808 nm and fitted into linear function. Two slopes, one from the reference of IR-26 (quantum yield 0.5% in 1,2-dichloroethane) and the other from BTPE-NH$_2$@F127 (in PBS) or BTPE-NH$_2$ (in PBS containing 5% DMSO), were adopted for the calculation of the quantum yield. The fluorescence quantum yield was calculated as follows:

$$QY_{sample} = QY_{ref} \frac{Slope_{sample}}{Slope_{ref}} \left( \frac{n_{sample}}{n_{ref}} \right)^2 \qquad (4)$$

where $QY_{ref}$ represents the fluorescence quantum yield of IR-26 in dichloroethane, $QY_{sample}$ represents the fluorescence quantum yield of BTPE-NH$_2$@F127 in PBS or BTPE-NH$_2$ in PBS containing 5% DMSO, $n_{sample}$ denotes the refractive index of PBS solution or PBS containing 5% DMSO, and $n_{ref}$ represents the refractive index of 1,2-dichloroethane.

**Optical response of the nanoprobe BTPE-NO$_2$@F127 in solution**. The absorbance, optoacoustic and fluorescence signal changes of the nanoprobe BTPE-NO$_2$@F127 upon the addition of varied amounts of H$_2$O$_2$ (final BTPE-NO$_2$@F127 nanoparticle concentration: 350 µg·mL$^{-1}$, namely BTPE-NO$_2$ 32.6 µg·mL$^{-1}$) in PBS (10 mM, pH 7.4) at 37 °C were recorded after 90 min upon mixing. For time-dependent experiments, the solutions were kept at 37 °C for different time periods before spectral measurements. For selectivity experiments, H$_2$O$_2$ and/or other substances were added into BTPE-NO$_2$@F127 dispersions in PBS (10 mM, pH 7.4). Afterwards, optoacoustic and fluorescence signal intensities of the dispersions were recorded. The NIR-II fluorescence emission spectra were measured with the excitation of the 808 nm laser irradiation at 80 mW cm$^{-2}$.

**Cell culture, cytotoxicity studies and cell imaging**. Murine macrophage cell line (RAW264.7) was obtained from KeyGen Biology Co. Ltd (Nanjing, China). RAW264.7 cells were cultured in Dulbecco's modified eagle medium (DMEM) supplemented with 10% fetal bovine serum (FBS) and 1% penicillin and streptomycin at 37 °C in 5% CO$_2$ atmosphere. The viabilities of RAW264.7 cells that were or were not exposed to BTPE-NO$_2$@F127 were evaluated by MTT assay according to ISO 10993–5. Briefly, the cells were seeded in 96-well plates at 5000 cells/well and cultured for 24 h. Afterwards, the cells were washed with PBS for three times and incubated with different concentrations of BTPE-NO$_2$@F127 (0, 50, 100, 200, 300 and 500 µg mL$^{-1}$) in the medium for another 24 h. Then, the new medium containing 0.5 mg mL$^{-1}$ MTT was added to each well of the 96-well assay plate and incubated for additional 4 h. Finally, the medium was washed three times with PBS and replaced with DMSO (150 µL) to dissolve the precipitates. The absorbance was later measured with a Thermo MK3 ELISA reader at 570 nm to estimate the viability of cells.

For cell imaging, RAW264.7 cells ($1 \times 10^6$) were seeded in the 6-well plates and incubated for 24 h. After cells adhered to the plates, the cells were washed with PBS and then incubated in DMEM with 10% FBS containing different concentrations (0, 50, 100, 150, 200 µM) of H$_2$O$_2$ for 1 h. Afterwards, the nanoprobe (400 µg·mL$^{-1}$, BTPE-NO$_2$ 37.2 µg mL$^{-1}$) was added and incubated for additional 4 h. Then, the cells were washed, trypsinized, centrifuged, suspended in PBS and subjected to NIR-II fluorescence imaging using NIR-II in Vivo Imaging System (Series II 808/900–1700, Suzhou NIR-Optics Technologies Co., Ltd.) (excitation: 808 nm laser with 50 mW cm$^{-2}$, emission filter: 900–1700) as well as MSOT imaging (excitation: 680 nm). For cell experiments in MSOT imaging, the control (the cells without incubation with H$_2$O$_2$) or the treated cells suspended in PBS was fully filled in commercial Wilmad NMR tubes respectively and then fixed on the holder of the imaging instrument.

**Relative optoacoustic intensity**. The relative optoacoustic (OA) intensity was used to reflect the response of the nanoprobe toward varied concentrations of H$_2$O$_2$. It was calculated according to the equation:

$$\text{Relative OA intensity} = [(OA_{680})_{H_2O_2} - (OA_{680})_{control}]/(OA_{680})_{control} \qquad (5)$$

**Animal experiments**. The BALB/C mice (male, 6–7 weeks old) and BALB/C mice (female, 6–7 weeks old) were purchased from Guangdong Medical Laboratory Animal Center (GDMLAC) and kindly kept in the Laboratory Animal Center of South China Agricultural University. The in vivo experiments were approved and conducted in compliance with the regulations of Ethics Committee of Laboratory Animal Center of South China Agricultural University (Approval No.: 2020-d087). The study was performed in accordance with the Regulations on the

Administration of Laboratory Animals of Guangdong Province and the Regulations on the Management of Laboratory Animals of China. Mice were housed in sterile cages within laminar airflow hoods at 24 °C and 45–65% humidity in a specific pathogen-free room with a 12 h light/12 h dark schedule, and fed autoclaved chow and water ad libitum. Mice were divided randomly to establish different animal models and support subsequent experimental studies. In the case of lethal experimental procedures, mice would be euthanized by exposure to carbon dioxide gas. Before the imaging experiments, depilatory cream was used for hair removal in mice.

**Mouse model of interstitial cystitis**. The interstitial cystitis mouse model was established according to the previous reports[64]. Briefly, female mice (7–8 weeks old) were randomly divided into groups with five mice per group, considering sufficient replications of results along with reduction of animal number. The animals were treated with varied dosages of cyclophosphamide (CYP, 75 mg kg$^{-1}$ and 150 mg kg$^{-1}$) or isovolumic saline (control group) via intraperitoneal administration. After 24 h, the mice underwent imaging experiments post intravesical injection of the nanoprobe.

**Trazodone-induced liver injury mouse model**. Male mice (7–8 weeks old) were randomly divided into groups with five mice per group, balancing sufficient replication of results with a reduction in animal number. For trazodone-induced liver injury, groups of male mice were treated with varied dosages of trazodone hydrochloride (50, 100 and 200 mg kg$^{-1}$) or isovolumic saline (control group) via intraperitoneal administration for three consecutive days. 6 h later after the last injection, the imaging was performed at varied time periods post i.v. injection of the nanoprobe.

**Liver ischemia/reperfusion injury mouse model**. The liver ischemia/reperfusion injury mouse model was established according to previous report[65]. The male BALB/C mice (7–8 weeks old) were randomly divided into three groups, namely, the sham-surgery group ($n = 5$), the short ischemia group ($n = 5$) and the long ischemia group ($n = 5$). The mice in the sham-surgery group were given a midline laparotomy surgery without ischemia. The mice in the short ischemia group were given a midline laparotomy surgery, and ischemia was induced by clamping the hepatic artery and portal vein for 30 min followed by reperfusion for 24 h. The mice in the long ischemia group were given a midline laparotomy surgery, and ischemia was induced by clamping the hepatic artery and portal vein for 60 min followed by reperfusion for 24 h. During the midline laparotomy surgery, the mice were anesthetized by inhaling 1% isoflurane and the ischemia was performed using atraumatic vascular clamp. After 30 min or 60 min of ischemia, the clamp was removed, followed by suturing the abdominal cavity layer by layer and then reperfusion for 24 h.

**In vivo NIR-II fluorescent imaging**. The mice were placed in the supine posture. For imaging of IC syndrome, 1% isoflurane was delivered with nose cone for mouse anesthesia, and the mice were intravesically injected with the nanoprobe BTPE-NO$_2$@F127 dispersion (in PBS, BTPE-NO$_2$ 2.3 mg kg$^{-1}$) into the bladder via a 24-gauge catheter. After 15 min, the catheter was pulled out, the mice were anesthetized with continuous isoflurane and imaged. For imaging of trazodone-induced liver injury and liver ischemia/reperfusion injury, the mice were treated by i.v. injection of the nanoprobe BTPE-NO$_2$@F127 dispersion (equivalent to 9.1 mg kg$^{-1}$ of BTPE-NO$_2$). The excitation light was provided by an 808 nm laser with a power density at the imaging plane of 50 mW cm$^2$.

**Multispectral optoacoustic tomography imaging**. All in vitro phantom and in vivo mouse optoacoustic imaging experiments were carried out on a multispectral optoacoustic tomographic imaging system.

For phantom experiments, the test solution or the control (PBS) was fully filled in commercial Wilmad NMR tubes respectively and then fixed on the holder of the imaging instrument. In vitro optoacoustic images of the nanoprobe BTPE-NO$_2$@F127 in the presence of H$_2$O$_2$ of varied concentrations were acquired (signal at 680 nm).

For in vivo MSOT imaging experiments of the IC mouse model, the mice were anesthetized by 1% isoflurane delivered with nose cone. After intravesical injection with the nanoprobe BTPE-NO$_2$@F127 dispersion (in PBS, BTPE-NO$_2$ 2.3 mg kg$^{-1}$) into the bladder through a 24-gauge catheter, the mice were placed in the prone position in a water bath at 34 °C, and anesthesia and oxygen were supplied using a breathing mask. The following imaging wavelengths were selected by considering the major turning points in the absorption spectra of BTPE-NH$_2$ and hemoglobin: 680 nm, 700 nm, 730 nm, 760 nm, 800 nm, 850 nm (background) and 900 nm. For each wavelength, 10 individual frames were recorded. The z-stack of the cross-sectional images created the orthogonal-view three-dimensional (3D) images. As for in vivo MSOT imaging experiments of the trazodone-induced liver injury mouse model and the liver ischemia/reperfusion injury mouse model, the nanoprobe BTPE-NO$_2$@F127 dispersion (in PBS, equivalent to 9.1 mg kg$^{-1}$ of BTPE-NO$_2$) was i.v. injected and imaging was performed. Similarly, the trunk of the mice covering whole liver region with a step size of 0.3 mm was selected for MSOT imaging. Subsequently, guided ICA spectral unmixing was used to separate

signals coming from BTPE-NH$_2$ (the activated probe) and those from endogenous photo-absorbing elements in the body (e.g., hemoglobin). As for in vivo mouse imaging experiments, five mice were tested for every group.

**Tissue histological evaluation**. For the study of the IC model, 24 h after cyclophosphamide injection, the mice from different groups were euthanized and the bladder was excised. The bladder tissues were embedded in paraffin and sectioned to 4 μm for hematoxylin and eosin (H&E) staining.

For the study of the trazodone-induced liver injury model, 6 h after the trazodone injection (for the last time) on the third day, the mice from different groups were euthanized and the liver was excised for H&E staining.

For the study of the liver ischemia/reperfusion injury model, after reperfusion for 24 h, the livers from different groups were excised for H&E staining.

For biosafety study, healthy mice were intravenously injected with the nanoprobe BTPE-NO$_2$@F127 dispersion (BTPE-NO$_2$ 9.1 mg kg$^{-1}$) or isovolumic saline. After 24 h, the mice were euthanized and the corresponding organs (heart, liver, spleen, lung and kidney) were excised, and the organ tissues were embedded in paraffin and sectioned to 4 μm for H&E staining.

**Serum biochemistry assessment**. The serum was obtained from different groups and the indexes including alanine aminotransferase (ALT) and aspartate transaminase (AST) levels were measured by using Elisa kits.

**Body weight measurements**. For evaluating in vivo toxicity of BTPE-NO$_2$@F127 nanoprobe, body weights of mice were measured. For the body weight measurements, mice ($n = 5$ per group, 7 weeks old) were i.v. injected with the nanoprobe dispersion (BTPE-NO$_2$ 9.1 mg·kg$^{-1}$) or with isovolumic saline. The body weights of these mice were recoded for 7 days.

**Statistical analysis**. All experiments were repeated at least three times. Quantitative data were expressed as mean ± standard error (SD). NMR spectra were analyzed using Mestre Nova LITE v14.0.0.0–23239 software (Mestre lab Research S.L.). Statistical calculations were performed using OriginPro 2018 (64 bit) SR1 b9.5.1.195. Data analysis of absorption spectra and fluorescence spectra was performed using OriginPro 2018 (64 bit) SR1 b9.5.1.195.

**Reporting summary**. Further information on research design is available in the Nature Research Reporting Summary linked to this article.

## Data availability

All the other data supporting the findings of this study are available within the article and its supplementary information files and from the corresponding author upon reasonable request.

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

## Acknowledgements

This work was supported by the National Natural Science Foundation of China (No. 21788102 to S.W. and No. 21875069 to F.Z.), the Natural Science Foundation of Guangdong Province (No. 2016A030312002 to S.W.), and the Fund of Guangdong Provincial Key Laboratory of Luminescence from Molecular Aggregates (No. 2019B030301003 to S.W.). The work was also supported by the Singapore Agency for Science, Technology and Research (A*STAR) AME IRG grant (No. A20E5c0081 to Y.Z.) and the Singapore National Research Foundation Investigatorship (No. NRF-NRFI2018-03 to Y.Z.).

## Author contributions

Y.Z., S.W., F.Z., and J.C. conceived the project. J.C. and L.C. designed the chemical synthetic route. J.C., L.C., Y.W. and Y.F. conducted the experiments. Y.Z., S.W., F.Z., and J.C. wrote the manuscript. All the other co-authors contributed to data interpretation and revision of manuscript.

## Competing interests

The authors declare no competing interests.

## Additional information

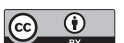

