## [Peer Review File · Nature Communications]

Reviewers' comments:

Reviewer #1 (Remarks to the Author):

Fluorescent imaging in the second biological window (emission in 1000 – 1700 nm, NIR-II) is of great significance and has a major impact on the field of preclinical and clinical applications, hence high-quality NIR-II fluorophores are urgently needed. In this manuscript, the authors present an activatable nanoprobe, which can circumvent the commonly-encountered limitations associated with NIR-II fluorophores with large and planar conjugation including poor water solubility and the accompanying fluorescence quenching due to aggregation in aqueous milieu and non-activatable response. The nanoprobe has been applied in several mouse models for biomarker detection and diseases imaging via NIR-II fluorescent imaging and MSOT imaging. Overall, the work is interesting and carries technological novelty, the manuscript is well presented with proper discussion, and the molecular-structure design strategy would offer inspirational information for devising other large-conjugation NIR-II chromophores for a variety of biological applications. Therefore, I would recommend its publication after the following minor issues are addressed:

1. Spectral properties including absorption spectra and fluorescence spectra can be adopted to further verify the reaction mechanism of the nanoprobe BTPE-NO₂@127 with H₂O₂. For this reason, the absorption spectra and fluorescence spectra of the product after the nanoprobe BTPE-NO₂@F127's reaction with H₂O₂ should be compared to those of BTPE-NH₂@F127 nanoparticles consisting of the theoretical reaction product-the dye BTPE-NH₂.
2. The spectral changes (e.g., absorption and fluorescence spectra) of the chromophore BTPE-NH₂ may be measured in different organic solvents (such as high-polar solvent DMSO and low-polar solvent DCM), so as to illustrate the solvent polarity on absorption and emission.
3. What's the detection limit of the nanoprobe BTPE-NO₂@127 toward H₂O₂?
4. What is the stability of the nanoprobe BTPE-NO₂@127 in PBS containing 10% FBS?
5. In the figure captions of Figure 2A and 2D, the excitation wavelength should be stated.

Reviewer #2 (Remarks to the Author):

The authors developed a H₂O₂ activatable NIR-II fluorophore (BTPE-NO₂) for NIR-II FL and PAI imaging of inflammatory diseases. The obtained BTPE-NO₂ could transformed into the chromophore BTPE-NH₂ and generating strong NIR-II emission and 680 nm photoacoustic signal after incubation with H₂O₂. Although this modality exhibited good performances for in vivo trazodone-induced liver injury, I/R liver injury, and interstitial 494 cystitis mouse models. However, the above three mouse models have been well studied in previous reports using NIR-II FL and PAI imaging probes. And this work did not shown any advantages (such as enhanced signal-to-background ratio) compared with previous reported probes. For these reasons, I therefore recommend against publication in the Nature Communications.

some minors:

1. The authors should provide evidence to demonstrate the stability of BTPE-NO₂@F127 after responding to H₂O₂, such as TEM image, DLS, and NIR-II FL.
2. The mass extinction coefficient of BTPE-NO₂ and BTPE-NH₂ should be provided.
3. The fluorescent quantum yields of BTPE-NH₂ and BTPE-NH₂@F127 should be reported, and also need to compare with other reported organic NIR-II FL materials.
4. The authors need to compare their NIR-II fluorescence and MSOT imaging results of liver I/R injury and liver injury with already reported literature of the same kind of disease models. What are the advantages of BTPE-NH₂@F127 compared with literature reported imaging probes?

Reviewer #3 (Remarks to the Author):

Imaging in the NIR-II region (1000-1700 nm) is a very important technique to enhance the spatial resolution and tissue penetration depth. In this manuscript, the authors reported H₂O₂ activatable nanoprobe (BTPE-NO₂@F127) based on a D-A-D backbone structure and demonstrated their feasibility for in vivo diagnosis. However, there are serious problems with the experimental data, as outlined in my comments to the authors.

1. There has been no evidence that 40 nm nanoparticles can be cleared to the bladder after IV injection. Therefore, it is hard to believe the signals from the intact form of BTPE-NO₂@F127 can be detected in the bladder of the interstitial cystitis mouse model. There is also a lack of in vivo data (PK, biodistribution, long-term clearances, and biochemical analysis, e.g., AST and ALT) in demonstrating potential in vivo diagnostic applications.
2. Since the authors used an 808 nm laser which does not match the extinction wavelength of BTPE-NO₂@F127, it is unclear whether the fluorescence signal intensity is restored due to the red-shift of the excitation wavelength or because of fluorescence dequenching after being cleaved by H₂O₂.
3. Showing the Tyndall phenomenon does not indicate the stability of BTPE-NO₂@F127 in size changes. Are there any changes in particle size or zeta potential for the nanoprobe BTPE-NO₂@F127 after incubation with H₂O₂?
4. Multispectral optoacoustic tomography imaging was not conducted in the NIR-II region.
5. The title should be changed to H₂O₂-Activable Nanoprobe.

Reviewer #1:

Fluorescent imaging in the second biological window (emission in 1000 – 1700 nm, NIR-II) is of great significance and has a major impact on the field of preclinical and clinical applications, hence high-quality NIR-II fluorophores are urgently needed. In this manuscript, the authors present an activatable nanoprobe, which can circumvent the commonly-encountered limitations associated with NIR-II fluorophores with large and planar conjugation including poor water solubility and the accompanying fluorescence quenching due to aggregation in aqueous milieu and non-activatable response. The nanoprobe has been applied in several mouse models for biomarker detection and diseases imaging via NIR-II fluorescent imaging and MSOT imaging. Overall, the work is interesting and carries technological novelty, the manuscript is well presented with proper discussion, and the molecular-structure design strategy would offer inspirational information for devising other large-conjugation NIR-II chromophores for a variety of biological applications. Therefore, I would recommend its publication after the following minor issues are addressed:

Our response: We thank Reviewer #1 for giving the positive comments for this work. We have carefully revised the manuscript based on the comments and suggestions.

1. Spectral properties including absorption spectra and fluorescence spectra can be adopted to further verify the reaction mechanism of the nanoprobe BTPE-NO₂@127 with H₂O₂. For this reason, the absorption spectra and fluorescence spectra of the product after the nanoprobe BTPE-NO₂@F127's reaction with H₂O₂ should be compared to those of BTPE-NH₂@F127 nanoparticles consisting of the theoretical reaction product-the dye BTPE-NH₂.

Our response: The comparison of absorption spectra and fluorescence spectra for BTPE-NH₂@F127 and BTPE-NO₂@F127 after response to H₂O₂ has been conducted, the data have been added as Figure S24B,C, and the related description has been added on Page 12 in the revised manuscript.

2. The spectral changes (e.g., absorption and fluorescence spectra) of the chromophore BTPE-NH₂ may be measured in different organic solvents (such as high-polar solvent DMSO and low polar solvent DCM), so as to illustrate the solvent polarity on absorption and emission.

Our response: The spectral changes of BTPE-NH₂ have been measured in different organic solvents (such as high-polar solvents DMSO and DMF, as well as the low polar solvents toluene and dichloromethane), the data have been provided as Figure S21C-D, and the related description has been added on Page 11 in the revised manuscript.

3. What's the detection limit of the nanoprobe BTPE-NO₂@127 toward H₂O₂?

Our response: The detection limit of BTPE-NO₂@127 toward H₂O₂ has been determined, the data have been provided as Figure S21B, and the related description has been added on Page 11 in

the revised manuscript.

4. What is the stability of the nanoprobe BTPE-NO₂@127 in PBS containing 10% FBS?

Our response: The stability of the nanoprobe BTPE-NO₂@127 in PBS containing 10% FBS has been measured, and the data have been provided in Figure S18.

5. In the figure captions of Figure 2A and 2D, the excitation wavelength should be stated.

Our response: In the figure captions of Figure 2A and 2D, the excitation wavelength has been stated.

Reviewer #2:

The authors developed a H₂O₂ activatable NIR-II fluorophore (BTPE-NO₂) for NIR-II FL and PAI imaging of inflammatory diseases. The obtained BTPE-NO₂ could be transformed into the chromophore BTPE-NH₂ and generating strong NIR-II emission and 680 nm photoacoustic signal after incubation with H₂O₂. Although this modality exhibited good performances for in vivo trazodone-induced liver injury, I/R liver injury, and interstitial cystitis mouse models. However, the above three mouse models have been well studied in previous reports using NIR-II FL and PAI imaging probes. And this work did not show any advantages (such as enhanced signal-to-background ratio) compared with previous reported probes. For these reasons, I therefore recommend against publication in the Nature Communications.

Our response: We thank the reviewer for this comment, which along with his/her raised minor issues would help improve the overall quality of our manuscript. The I/R liver injury and interstitial cystitis mouse models have been established, while the trazodone-induced liver injury has been discovered clinically (e.g., *Am. J. Gastroenterol.* 2000, 95, 532-535; *Am. J. Psychiatry* 2014, 171, 404-415; *Ann. Pharmacother.* 2001, 35, 1559-1561) and rarely been studied on animal models. Our main purpose is not to study the establishment of these mouse models, instead we aim to develop an NIR-II activatable probe with the new molecular design strategy and use these mouse models to evaluate the probe's capability for detecting and imaging inflammatory disease sites via responding to the in-situ biomarker. We are sorry for not stating sufficiently clear on this point in the previous version of our manuscript. The related descriptions have been added on Page 6 in the revised manuscript.

In this revised manuscript, we have compared our probe with other reported NIR-II fluorescent probes and PAI probes. As shown in Tables S1-S3, it is clear that our probe outperforms other NIR-II and PAI probes in many ways (e.g., activatable response and signal-to-background ratio). In addition, the design principle of our probe is totally different to other reported probes, revealing its fundamental novelty. The related descriptions have been added on Page 24 in the revised manuscript.

some minors:

1. The authors should provide evidence to demonstrate the stability of BTPE-NO₂@F127 after responding to H₂O₂, such as TEM image, DLS, and NIR-II FL.

Our response: The stability of BTPE-NO₂@F127 after responding to H₂O₂ such as TEM image, DLS, and NIR-II FL has been measured, and the data (TEM image, DLS, and NIR-II FL) have been provided in Figure S19E,F. Related descriptions have been added on Pages 8-9 in this revised manuscript.

2. The mass extinction coefficient of BTPE-NO₂ and BTPE-NH₂ should be provided.

Our response: The mass extinction coefficient of BTPE-NO₂ and BTPE-NH₂ has been provided on

Page 12 in the revised manuscript.

3. The fluorescent quantum yields of BTPE-NH₂ and BTPE-NH₂@F127 should be reported, and also need to compare with other reported organic NIR-II FL materials.

Our response: The fluorescent quantum yields of BTPE-NH₂ and BTPE-NH₂@F127 have been determined and compared with other reported organic NIR-II FL materials, as shown in Table S1.

4. The authors need to compare their NIR-II fluorescence and MSOT imaging results of liver I/R injury and liver injury with already reported literature of the same kind of disease models. What are the advantages of BTPE-NH₂@F127 compared with literature reported imaging probes?

Our response: The imaging performance of probe BTPE-NO₂@F127 in the I/R liver injury, drug-induced liver injury and bladder disease mouse models has been compared with the literature-reported imaging probes, as shown in Tables S2 and S3. It is clear that our probe outperforms other NIR-II and PAI (optoacoustic imaging) probes. More importantly, this molecular design strategy could offer helpful insights for designing other large conjugation NIR-II chromophores for various biological applications.

Herein, we would like to further explain the significance of this work. The NIR-II fluorescent imaging (emission in 900–1700 nm) could enable real-time imaging at deeper depth with higher resolution for more accurate disease diagnosis, monitoring therapy course and understanding disease development. Among the NIR-II organic fluorescent dyes, benzothiadiazole-core dyes have aroused extensive attention because of their ease of extending fluorescent emission into the NIR-II range and excellent photostability. However, these fluorophores still exhibit some limitations associated with their large and planar conjugation, which include poor water solubility accompanied with fluorescence quenching in aqueous biological milieu due to aggregation. In addition, these fluorophores are generally inert probes, presenting “always-on” signals that constitute background noise. In contrast, the activatable probes produce signals only under a specific stimulus, and hence the detection or imaging by using activatable probes would have much higher sensitivity with negligible background noise, thus effectively avoiding false positive results.

Aiming to overcome these limitations, on one hand, we have incorporated molecular rotors with aggregation-induced emission (AIE) capability into the benzothiadiazole core structure to alleviate the π - π stacking between the benzothiadiazoles and afford the fluorophore with the AIE feature that generates enhanced fluorescence in aqueous milieu. By this way, the poor water-solubility disadvantage could be turned into an advantageous asset for devising NIR-II fluorophores on the strength of AIE. On the other hand, we have designed the responsive probe that can be achieved by the rational integration of two recognition moieties at both ends of the benzothiadiazole core. Thus, the molecular probe BTPE-NO₂ has been formed with two nitrophenyloxoacetamide units at both ends of the core as recognition moieties and fluorescence quenchers. The de novo nanoprobe named BTPE-NO₂@F127 has been developed

with an FDA-approved polymer Pluronic F127 being employed to encapsulate the molecular probe BTPE-NO₂. The pathological levels of H₂O₂ in the disease sites could cleave the nitrophenyloxoacetamide groups and activate the probe, thereby generating strong fluorescent emission (950-1200 nm) and optoacoustic signals for dual-mode imaging of several inflammatory diseases and locating inflammatory foci.

Therefore, we believe that the present study unleashes the potential of benzothiadiazole-core fluorophores for “turn-on” NIR-II fluorescent and optoacoustic imaging in biological applications, and the molecular design strategy could offer insights for designing other large conjugation NIR-II fluorophores for various biological applications. Thanks for your further consideration.

Reviewer #3:

Imaging in the NIR-II region (1000-1700 nm) is a very important technique to enhance the spatial resolution and tissue penetration depth. In this manuscript, the authors reported H₂O₂ activatable nanoprobe (BTPE-NO₂@F127) based on a D-A-D backbone structure and demonstrated their feasibility for in vivo diagnosis. However, there are serious problems with the experimental data, as outlined in my comments to the authors.

Our response: We thank the reviewer for the useful comments, and we have carefully revised the manuscript based on the suggestions.

1. There has been no evidence that 40 nm nanoparticles can be cleared to the bladder after IV injection. Therefore, it is hard to believe the signals from the intact form of BTPE-NO₂@F127 can be detected in the bladder of the interstitial cystitis mouse model. There is also a lack of in vivo data (PK, biodistribution, long-term clearances, and biochemical analysis, e.g., AST and ALT) in demonstrating potential in vivo diagnostic applications.

Our response: Thanks for the comments. The probe is actually **intravesically** (not intravenously) injected into the mouse bladder. We use an interstitial cystitis mouse model to evaluate the probe's capability for detecting and imaging inflammatory diseases. The data concerning PK, biodistribution, long-term clearances, and biochemical analysis, e.g., AST and ALT have already been measured during the review period and provided in Figure S29, and the related descriptions have been added on Pages 13-14 in the revised manuscript.

2. Since the authors used an 808 nm laser which does not match the extinction wavelength of BTPE-NO₂@F127, it is unclear whether the fluorescence signal intensity is restored due to the red-shift of the excitation wavelength or because of fluorescence dequenching after being cleaved by H₂O₂.

Our response: To prove the origin of the increased fluorescence, the fluorescence spectra before and after the response of the probe BTPE-NO₂@F127 to hydrogen peroxide have been measured using 808 nm laser as excitation light (Figure S19A). Additionally, the fluorescence spectra of the BTPE-NH₂@F127 and BTPE-NO₂@F127 have also been measured using 615 nm (maximum absorption of BTPE-NO₂@F127) as excitation light (Figure S19C,D). From these data, it is clear that the increased fluorescence intensity is indeed the result of the probe's response toward H₂O₂. The related descriptions have been added on Pages 8-9 in the revised manuscript.

3. Showing the Tyndall phenomenon does not indicate the stability of BTPE-NO₂@F127 in size changes. Are there any changes in particle size or zeta potential for the nanoprobe BTPE-NO₂@F127 after incubation with H₂O₂?

Our response: The changes in particle size or zeta potential for the nanoprobe BTPE-NO₂@F127 after incubation with H₂O₂ have been measured, and the data have been provided in Figure S19E.

Related descriptions have been added on Pages 8-9 in this revised manuscript.

4. Multispectral optoacoustic tomography imaging was not conducted in the NIR-II region.

Our response: Indeed, MSOT imaging is not conducted in the NIR-II region. To avoid possible confusion, the manuscript title has been modified as “A H₂O₂-Activatable Nanoprobe for Diagnosing Interstitial Cystitis and Liver Ischemia-Reperfusion Injury via Multispectral Optoacoustic Tomography and NIR-II Fluorescent Imaging”.

5. The title should be changed to H₂O₂-Activatable Nanoprobe.

Our response: As suggested, the manuscript title has been modified as “A H₂O₂-Activatable Nanoprobe for Diagnosing Interstitial Cystitis and Liver Ischemia-Reperfusion Injury via Multispectral Optoacoustic Tomography and NIR-II Fluorescent Imaging”.

REVIEWERS' COMMENTS

Reviewer #1 (Remarks to the Author):

The authors responded all the questions. I am satisfied with the revised version. It should be accepted.

Reviewer #2 (Remarks to the Author):

The quality of the revised manuscript has been greatly improved. Thus, I recommend it for publication in this journal.

Reviewer #3 (Remarks to the Author):

The authors have done a fair job of answering the questions and have been making edits suggested. However, the experimental details for the measurement of fluorescent quantum yields of BTPE-NH₂ and BTPE-NH₂@F127 should be included. With the method, this could be accepted.

Point-by-point responses to the comments of reviewers

Reviewer #1:

The authors responded all the questions. I am satisfied with the revised version. It should be accepted.

Our response: We thank the reviewer for the positive comments.

Reviewer #2:

The quality of the revised manuscript has been greatly improved. Thus, I recommend it for publication in this journal.

Our response: We thank the reviewer for the positive comments.

Reviewer #3:

The authors have done a fair job of answering the questions and have been making edits suggested. However, the experimental details for the measurement of fluorescent quantum yields of BTPE-NH₂ and BTPE-NH₂@F127 should be included. With the method, this could be accepted.

Our response: We thank the reviewer for the positive comment and suggestion. We have added the experimental details for the fluorescent quantum yield measurements of BTPE-NH₂ and BTPE-NH₂@F127 in the Methods section in the revised manuscript.